# Subgaussian and Differentiable Importance Sampling for Off-Policy Evaluation and Learning

**Alberto Maria Metelli**
DEIB, Politecnico di Milano
albertomaria.metelli@polimi.it

**Alessio Russo**
DEIB, Politecnico di Milano
alessio.russo@polimi.it

**Marcello Restelli**
DEIB, Politecnico di Milano
marcello.restelli@polimi.it

## Abstract

Importance Sampling (IS) is a widely used building block for a large variety of off-policy estimation and learning algorithms. However, empirical and theoretical studies have progressively shown that vanilla IS leads to poor estimations whenever the behavioral and target policies are too dissimilar. In this paper, we analyze the theoretical properties of the IS estimator by deriving a novel anticoncentration bound that formalizes the intuition behind its undesired behavior. Then, we propose a new class of IS transformations, based on the notion of power mean. To the best of our knowledge, the resulting estimator is the first to achieve, under certain conditions, two key properties: (i) it displays a subgaussian concentration rate; (ii) it preserves the differentiability in the target distribution. Finally, we provide numerical simulations on both synthetic examples and contextual bandits, in comparison with off-policy evaluation and learning baselines.

## 1 Introduction

The availability of historically collected data is a common scenario in many real-world decision-making problems, including medical treatments [e.g., 17, 67], recommendation systems [e.g., 33, 16], personalized advertising [e.g., 3, 60], finance [e.g., 43], and industrial robot control [e.g., 27, 26]. Historical data can be leveraged to address two classes of problems. First, given data collected with a *behavioral* policy, we want to estimate the performance of a different *target* policy. This problem is known as *off-policy evaluation* [Off-PE, 21]. Second, we want to employ the available data to improve the performance of a baseline policy. This latter problem is named *off-policy learning* [Off-PL 14]. Off-policy methods are studied by both the *reinforcement learning* [RL, 58] and *contextual multi-armed bandit* [CMAB, 30] communities. Given its intrinsic simplicity compared to RL, off-policy methods are nowadays well understood in the CMAB framework [e.g., 44, 1, 14, 64]. Among them, the *doubly robust* estimator [DR, 14] is one of the most promising off-policy methods for CMABs. DR combines a *direct method* (DM), in which the reward is estimated from historical data via regression, with an *importance sampling* [IS, 46] control variate.

IS plays a crucial role in the off-policy methods and counterfactual reasoning. However, IS tends to exhibit problematic behavior for general distributions. This is formalized by its *heavy-tailed* properties [40], which prevent the application of exponential concentration bounds [4]. To cope with this issue, typically, corrections are performed on the importance weight including *truncation* [23] and *self-normalization* [SN, 46], among the most popular. Significant results have recently been derived for both techniques [47, 29, 39]. Nevertheless, we believe that the widespread use of IS calls for a better theoretical understanding of its properties and for the design of principled weight corrections.

35th Conference on Neural Information Processing Systems (NeurIPS 2021).

Defining the desirable properties of an off-policy estimator is a non-trivial task. Some works employed the *mean squared error* (MSE) as an index of the estimator quality [34, 64]. However, controlling the MSE, while effectively capturing the bias-variance trade-off, does not provide any guarantee on the concentration properties of the estimator, which might still display a heavy-tailed behavior [37]. For this reason, we believe that a more suitable approach is to require that the estimator deviations concentrate at a *subgaussian* rate [12]. Subgaussianity implicitly controls the tail behavior and leads to tight exponential concentration inequalities. Unlike MSE, the probabilistic requirements are non-asymptotic (finite-sample), from which guarantees on the MSE can be derived. While subgaussianity can be considered a satisfactory requirement for Off-PE, additional properties are advisable when switching to Off-PL. In particular, the *differentiability* w.r.t. the target policy parameters is desirable whenever Off-PL is carried out via gradient optimization. For instance, weight truncation, as presented in [47], allows achieving subgaussianity but leads to a non-differentiable objective. Consequently, the optimization phase requires additional care, which sometimes leads to computationally heavy discretizations [47]. On the contrary, the SN estimator is differentiable in the target policy, but fails to achieve subgaussian concentration for general distributions.

In this paper, we take a step towards a better understanding of IS. After having introduced the necessary background (Section 2), we derive an anticoncentration bound for the mean estimation with vanilla IS. We show that polynomial concentration (Chebychev's inequality) is tight in this setting (Section 3). This result formalizes the intuition behind the undesired behavior of these estimators for general distributions. Hence, we propose a class of importance weight corrections, based on the notion of power mean (Section 4). The rationale behind these corrections is to "smoothly shrink" the weights towards the mean, with different intensities. In this way, we mitigate the heavy-tailed behavior and, in the meantime, we exert control over the induced bias. Then, we derive bounds on the bias and variance that allow obtaining an exponential concentration inequality and, under certain conditions, subgaussian concentration (Section 5). Furthermore, the smooth transformation allows preserving the differentiability in the target policy, unlike some existing transformations, like weight truncation. To the best of our knowledge, this is the first IS correction that preserves the differentiablity and is proved to be subgaussian. This correction, however, requires knowledge of a distributional divergence between the target and behavioral policies, which may be unknown or difficult to compute. To this end, we introduce an approach to empirically estimate the correction parameter, preserving the desirable concentration properties (Section 6). After providing a comparative review of the literature (Section 7), we present an experimental study comparing our approach with traditional and modern off-policy baselines on synthetic domains and in the CMAB framework (Section 8). The proofs of the results presented in the main paper can be found in Appendix A. A preliminary version of this work was presented at the "Workshop on Reinforcement Learning Theory" of ICML 2021 [42].[1]

## 2 Preliminaries

We start introducing the background about probability, importance sampling and contextual bandits.

**Probability** We denote with $\mathscr{P}(\mathcal{Y})$ the set of probability measures over a $(\mathcal{Y}, \mathfrak{F}_{\mathcal{Y}})$. Let $P \in \mathscr{P}(\mathcal{Y})$, $f : \mathcal{Y} \to \mathbb{R}$ be a function, and $\overline{\mu}_n$ be an estimator for the mean $\mu = \mathbb{E}_{y \sim P}[f(y)]$ obtained with $n$ i.i.d. samples. Suppose that with probability $1 - \delta$ it holds that $|\overline{\mu}_n - \mu| \leqslant \sqrt{g(n, \delta)}$. For $\beta > 0$, we say that $\overline{\mu}_n$ admits: (i) *polynomial* concentration if $g(n, \delta) = \mathcal{O}(1/(n\delta)^\beta)$; (ii) *exponential* concentration if $g(\delta) = \mathcal{O}((\log(1/\delta)/n)^\beta)$; (iii) *subgaussian* concentration if (ii) holds with $\beta = 1$ [37]. These cases correspond to Chebyshev's, Bernstein's, and Höeffding's inequalities respectively [4].

**Importance Sampling** Let $P, Q \in \mathscr{P}(\mathcal{Y})$ admitting $p$ and $q$ as density functions, if $P \ll Q$, i.e., $P$ is absolutely continuous w.r.t. $Q$, for any $\alpha \in (1, 2]$, we introduce the integral: $I_\alpha(P\|Q) = \int_{\mathcal{Y}} p(y)^\alpha q(y)^{1-\alpha} \mathrm{d}y$. If $P = Q$ a.s. (almost surely) then $I_\alpha(P\|Q) = 1$. $I_\alpha(P\|Q)$ allows defining several divergences, like Rényi [51]: $(\alpha - 1)^{-1} \log I_\alpha(P\|Q)$. Let $f : \mathcal{Y} \to \mathbb{R}$ be a function, *(vanilla) importance sampling* [IS, 46] allows estimating the expectation of $f$ under the *target* distribution $P$, i.e., $\mu = \mathbb{E}_{y \sim P}[f(y)]$, using i.i.d. samples $\{y_i\}_{i \in [n]}$ collected with the *behavioral* distribution $Q$:

$$\widehat{\mu}_n = \frac{1}{n} \sum_{i \in [n]} \omega(y_i) f(y_i), \qquad \text{where} \qquad \omega(y) = \frac{p(y)}{q(y)}, \quad \forall y \in \mathcal{Y}.$$

---

[1] https://lyang36.github.io/icml2021_rltheory/camera_ready/7.pdf.

It is well-known that $\widehat{\mu}_n$ is unbiased, i.e., $\mathbb{E}_{y_i \sim Q}[\widehat{\mu}_n] = \mu$ [46]. If $f$ is bounded, the variance of the estimator can be upper-bounded as $\mathbb{V}\mathrm{ar}_{y_i \sim Q}[\widehat{\mu}_n] \leqslant \frac{1}{n}\|f\|_\infty I_2(P\|Q)$ [40]. More in general, the integral $I_\alpha(P\|Q)$ represents the $\alpha$-moment of the importance weight $\omega(y)$ under $Q$.

**Contextual Bandits**   A *contextual multi-armed bandit* [CMAB, 30] is represented by the tuple $\mathcal{C} = (\mathcal{X}, \mathcal{A}, \rho, p)$, where $\mathcal{X}$ is the set of *contexts*, $\mathcal{A}$ is the finite set of actions (or arms), $\rho \in \mathscr{P}(\mathcal{X})$ is the context distribution, and $p : \mathcal{X} \times \mathcal{A} \to \mathscr{P}(\mathbb{R})$ is the reward distribution. The agent's behavior is encoded by a *policy* $\pi : \mathcal{X} \to \mathscr{P}(\mathcal{A})$. At each round $t \in \mathbb{N}$, the agent observes a context $x_t \sim \rho$, plays an action $a_t \sim \pi(\cdot|x_t)$, gets the reward $r_t \sim p(\cdot|x_t, a_t)$ and the system moves on to the next round. The *value* of a policy $\pi$ is given by $v(\pi) = \int_{\mathcal{X}} \rho(x) \sum_{a \in \mathcal{A}} \pi(a|x) \int_{\mathbb{R}} p(r|x, a) r \mathrm{d}r \mathrm{d}x$. We denote with $r(x, a) = \int_{\mathbb{R}} p(r|x, a) r \mathrm{d}r$ the *reward function*. A policy $\pi^*$ is optimal if it maximizes the value function, i.e., $\pi^* \in \arg\max_{\pi \in \Pi} v(\pi)$, where $\Pi = \{\pi : \mathcal{X} \to \mathscr{P}(\mathcal{A})\}$ is the set of all policies.

Let $\mathcal{D} = \{(x_t, a_t, r_t)\}_{t \in [n]}$ be a dataset of samples collected in a CMAB with a behavioral policy $\pi_b \in \Pi$. The *off-policy evaluation* [Off-PE, 21] problem consists in estimating the value function $v(\pi_e)$ of a target policy $\pi_e \in \Pi$ using the samples in $\mathcal{D}$. The *off-policy learning* [Off-PL, 14] problem consists in estimating an optimal policy $\pi^* \in \Pi$ using the samples in $\mathcal{D}$. The simplest approach to address the Off-PE/Off-PL problem is to learn from $\mathcal{D}$ an estimate $\widehat{r}(x, a)$ of the reward function $r(x, a)$ via regression. This approach is known as *direct method* (DM) and its properties heavily depend on the quality of the estimate $\widehat{r}$. Another approach is to simply apply IS to reweight the samples of $\mathcal{D}$, leading to the *inverse propensity scoring* [IS, 17] estimator. The two approaches are combined in the *doubly-robust* [DR, 14] estimator, in which the DM estimate is corrected with an IS control variate to reduce the variance using the estimated reward $\widehat{r}$ (see also Table 12 in Appendix D).

## 3   Anticoncentration of Vanilla Importance Sampling

In this section, we analyze the intrinsic limitations of the vanilla IS. It is well-known that under the assumption that for some $\alpha \in (1, 2]$ the divergence $I_\alpha(P\|Q)$ is finite and $f$ is bounded, the vanilla IS estimator $\widehat{\mu}_n$ admits *polynomial* concentration, i.e., with probability at least $1 - \delta$:[2]

$$|\widehat{\mu}_n - \mu| \leqslant \|f\|_\infty \left( \frac{2^{2-\alpha} I_\alpha(P\|Q)}{\delta n^{\alpha-1}} \right)^{\frac{1}{\alpha}}. \tag{1}$$

We now show that the concentration in Equation (1) is tight, by deriving an anticoncentration bound for $|\widehat{\mu}_n - \mu|$; then, we discuss its implications and compare it with previous works.

**Theorem 3.1.** *There exist two distributions $P, Q \in \mathscr{P}(\mathcal{Y})$ with $P \ll Q$ and a bounded measurable function $f : \mathcal{Y} \to \mathbb{R}$ such that for every $\alpha \in (1, 2]$ and $\delta \in (0, e^{-1})$ if $n \geqslant \delta e \max\left\{ 1, (I_\alpha(P\|Q) - 1)^{\frac{1}{\alpha-1}} \right\}$, with probability at least $\delta$ it holds that:*

$$|\widehat{\mu}_n - \mu| \geqslant \|f\|_\infty \left( \frac{I_\alpha(P\|Q) - 1}{\delta n^{\alpha-1}} \right)^{\frac{1}{\alpha}} \left( 1 - \frac{e\delta}{n} \right)^{\frac{n-1}{\alpha}}.$$

First of all, we note the polynomial dependence on the confidence level $\delta$. The bound is vacuous when $I_\alpha(P\|Q) = 1$, i.e., when $P = Q$ a.s., since in an on-policy setting and, being the function $f$ bounded, subgaussian concentration bounds (like Höeffding's inequality) hold. In particular, for $\alpha = 2$, $n$ and $I_2(P\|Q)$ sufficiently large, the bound has order $\mathcal{O}\big(\sqrt{I_2(P\|Q)/(\delta n)}\big)$, matching Chebyshev's and the existing concentration inequalities for vanilla importance sampling [40, 41].

Our result is of independent interest and applies for general distributions. Previous works considered the MAB [34] and CMAB [64] settings proving *minimax* lower bounds in *mean squared error* (MSE) $\mathbb{E}_{y \sim Q}[(\widehat{\mu}_n - \mu)^2]$. These results differ from ours in several respects. First, we focus on a *specific estimator*, the vanilla one, while those result are minimax. Second, they provide lower bounds to the MSE, while we focus on the deviations in probability.[3] From our probabilistic result, it is immediate to derive an MSE guarantee (Corollary A.1 of Appendix A.1). Finally, they assume that the second moment of the importance weight $I_2(P\|Q)$ is finite, whereas our result allows considering scenarios in which only moments of order $\alpha < 2$ are finite.

---

[2]The original result [41, Theorem 2] was limited to $\alpha = 2$ and based on Cantelli's inequality which approaches Chebyshev's when $\delta \to 0$. See Theorem A.1 in Appendix A.1, for a proof of Equation (1).

[3]As noted in [36], when the estimator is not well-concentrated around its mean (e.g., in presence of heavy tails), the MSE is not adequate to capture the error and high-probability bounds are more advisable.

| $s$ | $\omega_{s,\lambda}(y)$ |
| --- | --- |
| $-\infty$ (minimum) | $\min\{\omega(y),1\}$ |
| $-1$ (harmonic) | $\dfrac{\omega(y)}{1-\lambda+\lambda\omega(y)}$ |
| $0$ (geometric) | $\omega(y)^{1-\lambda}$ |
| $1$ (arithmetic) | $(1-\lambda)\omega(y)+\lambda$ |

Table 1: Choices of $s$ for the $(\lambda,s)$-corrected importance weight of Definition 4.1.

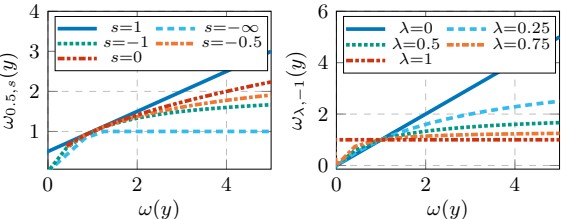

Figure 1: Examples of importance weight corrections of Definition 4.1 for fixed $\lambda$ (left) and fixed $s$ (right).

## 4 Power-Mean Correction of Importance Sampling

In this section, motivated by the negative result of Theorem 3.1, we look for a weight correction able to achieve subgaussian concentration. Specifically, we introduce a class of corrections based on the notion of *power mean* [6] and we study its properties. Let us start with the following definition.

**Definition 4.1.** *Let $P,Q\in\mathscr{P}(\mathcal{Y})$ be two probability distributions such that $P\ll Q$, for every $s\in[-\infty,\infty]$ and $\lambda\in[0,1]$, let $\omega(y)=p(y)/q(y)$, the $(\lambda,s)$-corrected* importance weight *is defined as:*

$$\omega_{\lambda,s}(y)=\Big((1-\lambda)\omega(y)^s+\lambda\Big)^{\frac{1}{s}},\quad\forall y\in\mathcal{Y}.$$

The correction can be seen as the weighted *power mean* with exponent $s$ between the vanilla importance weight $\omega(y)$ and 1 with weights $1-\lambda$ and $\lambda$ respectively.[4] We immediately notice that, regardless of the value of $s$, for $\lambda=0$, we reduce to the vanilla importance weight $\omega_{0,s}(y)=\omega(y)$ and for $\lambda=1$, we have identically $\omega_{1,s}(y)=1$. Furthermore, the correction is unbiased when $P=Q$ a.s. regardless $s$ and $\lambda$. Thus, the correction "smoothly interpolates" between the vanilla weight $\omega(y)$ and its mean under $Q$, i.e., 1. $s$ and $\lambda$ govern the "intensity" of the correction in a continuous way. Differently from the truncation [23], this transformation leads to a differentiable weight. Some specific choices of $s$ and $\lambda$ are reported in Table 1 and in Figure 1. The following result provides a preliminary characterization of the correction, independent of the properties of the two distributions.

**Lemma 4.1.** *Let $P,Q\in\mathscr{P}(\mathcal{Y})$ be two probability distributions with $P\ll Q$, then for every $\lambda\in[0,1]$ and $y\in\mathcal{Y}$ it holds that:*
  *(i) if $s\leqslant s'$ then $\omega_{\lambda,s}(y)\leqslant\omega_{\lambda,s'}(y)$;*
  *(ii) if $s<0$ then $\omega_{\lambda,s}(y)\leqslant\lambda^{\frac{1}{s}}$, otherwise if $s>0$ then $\omega_{\lambda,s}(y)\geqslant\lambda^{\frac{1}{s}}$;*
  *(iii) if $s<1$ then $\mathbb{E}_{y\sim Q}[\omega_{\lambda,s}(y)]\leqslant1$, otherwise if $s>1$ then $\mathbb{E}_{y\sim Q}[\omega_{\lambda,s}(y)]\geqslant1$.*

From point *(ii)* we observe that the corrected weight is bounded from below when $s>0$ and bounded from above when $s<0$. It is well-known that the inconvenient behavior of IS derives from the heavy-tailed properties [40]. Thus, the *arithmetic* correction ($s=1$) performs just a convex combination between the vanilla weight and 1, having no effect on the tail properties. Any correction with $s>1$ increases the weight value, making the tail even heavier. So, if we are looking for subgaussianity, we should restrict our attention to $s<0$, which leads to lighter tails or even bounded weights.

## 5 Subgaussian and Differentiable Importance Sampling

In this section, we focus on the *harmonic* correction ($s=-1$), which leads to a weight of the form:[5] $\omega_{\lambda,-1}(y)=\frac{\omega(y)}{1-\lambda+\lambda\omega(y)}$. We start analyzing the bias and variance of this class of estimators. Then, we provide an exponential concentration inequality that, under certain circumstances, results to be subgaussian. Finally, we show that the resulting estimator is differentiable in the target distribution. To lighten the notation we neglect the $-1$ subscript, abbreviating $\widehat{\mu}_\lambda=\widehat{\mu}_{\lambda,-1}$.

**Bias and Variance** We now derive bounds for the bias and the variance induced by the $(\lambda,-1)$-corrected importance weight.

---

[4]For $s\in\{-\infty,0,\infty\}$ the power mean is defined as a limit.
[5]The choice of $s=-1$ is mainly for analytical convenience and, as we shall see, it already allows enforcing the desired properties. We leave investigating the other values of $s$ for future work.

**Lemma 5.1.** *Let $P, Q \in \mathscr{P}(\mathcal{Y})$ be two probability distributions with $P \ll Q$. For every $\lambda \in [0,1]$, the bias and variance of the $(\lambda, -1)$-corrected importance weight can be bounded for every $\alpha \in (1,2]$ as:*

$$\left| \underset{y \sim Q}{\mathbb{E}} [\widehat{\mu}_{n,\lambda}] - \mu \right| \leqslant \|f\|_\infty \lambda^{\alpha-1} I_\alpha(P\|Q), \qquad \underset{y_i \sim Q}{\mathbb{V}\mathrm{ar}} [\widehat{\mu}_{n,\lambda}] \leqslant \frac{\|f\|_\infty^2}{n\lambda^{2-\alpha}} I_\alpha(P\|Q).$$

As expected, the bias is zero for $\lambda = 0$; it increases with $\lambda$ and with the divergence term $I_\alpha(P\|Q)$. Indeed, we already observed that the bias is null when $P = Q$ a.s.. In particular, for $\alpha = 2$, the bound becomes $\|f\|_\infty \lambda I_2(P\|Q)$. Instead, the variance bound decreases in $\lambda$ and increases with the divergence $I_\alpha(P\|Q)$. For $\alpha = 2$, we obtain the bound $\frac{1}{n}\|f\|_\infty^2 I_2(P\|Q)$. Note that when $P = Q$ a.s., we recover $\frac{1}{n}\|f\|_\infty^2$, which is the Popoviciu's inequality for the variance [50]. Thus, our weight correction allows controlling bias and variance even when $I_2(P\|Q) = \infty$, i.e., when the vanilla IS estimator might have infinite variance. Indeed, our transformed estimator has finite variance provided that there exists $\alpha \in (1,2)$ so that $I_\alpha(P\|Q) < \infty$. Tighter (but less intelligible) bounds on bias and variance are reported in Appendix A.3.

**Concentration Inequality** We are now ready to derive the core theoretical result. We prove an exponential concentration inequality for the $(\lambda, -1)$-corrected IS estimator and we show that, if $I_2(P\|Q)$ is finite, we are able to achieve subgaussian concentration.[6]

**Theorem 5.1.** *Let $P, Q \in \mathscr{P}(\mathcal{Y})$ be two probability distributions such that $P \ll Q$. For every $\alpha \in (1,2]$ and $\delta \in (0,1)$, if we select $\lambda = \lambda_\alpha^*$ then, with probability at least $1 - \delta$ it holds that:*

$$\widehat{\mu}_{n,\lambda_\alpha^*} - \mu \leqslant \|f\|_\infty (2 + \sqrt{3}) \left( \frac{2 I_\alpha(P\|Q)^{\frac{1}{\alpha-1}} \log \frac{1}{\delta}}{3(\alpha-1)^2 n} \right)^{1 - \frac{1}{\alpha}}, \quad with \quad \lambda_\alpha^* = \left( \frac{2 \log \frac{1}{\delta}}{3(\alpha-1)^2 I_\alpha(P\|Q) n} \right)^{\frac{1}{\alpha}}.$$

We immediately notice that the dependence on the confidence level $\delta$ is the one typical of exponential concentration for every $\alpha \in (1,2]$. In particular, we observe that the bound is subgaussian when $\alpha = 2$, requiring that $I_2(P\|Q) < \infty$. Recalling that $I_2(P\|Q)$ governs the variance of the estimator, this result is in line with the general theory of estimators for which the existence of the variance is an unavoidable requirement to achieve subgaussian concentration [12]. Specifically, for $\alpha = 2$ the optimal value of the parameter is $\lambda_2^* = \sqrt{(2\log(1/\delta))/(3 I_2(P\|Q) n)}$, leading to the bound:

$$\widehat{\mu}_{n,\lambda_2^*} - \mu \leqslant \|f\|_\infty (2 + \sqrt{3}) \sqrt{\frac{2 I_2(P\|Q) \log \frac{1}{\delta}}{3n}}. \tag{2}$$

A tighter bound, based on a different choice of the correction parameter $\lambda_\alpha^{**}$ is derived in Appendix A.3 and it is omitted here for clarity of presentation.

**Differentiability** As we have already observed, our weight correction, differently from truncation, is smooth and, thus, differentiable in the target policy. We now focus on the properties of the *gradient* of the $(\lambda, -1)$-corrected estimator and, to this purpose, we constrain the target distribution to belong to a parametric space differentiable distributions $\mathcal{P}_\Theta = \{ P_{\boldsymbol{\theta}} \in \mathscr{P}(\mathcal{Y}) : \boldsymbol{\theta} \in \Theta \}$, where $\Theta \subseteq \mathbb{R}^d$. The gradient of the corrected weight $\omega_\lambda$ w.r.t. the target policy parameters $\boldsymbol{\theta}$ is given by:

$$\nabla_{\boldsymbol{\theta}} \omega_\lambda(y) = \nabla_{\boldsymbol{\theta}} \frac{p_{\boldsymbol{\theta}}(y)}{q(y)} = \frac{(1-\lambda)\omega(y)}{(1 - \lambda + \lambda\omega(y))^2} \nabla_{\boldsymbol{\theta}} \log p_{\boldsymbol{\theta}}(y), \quad \forall y \in \mathcal{Y}.$$

In particular, it can be proved that $\|\nabla_{\boldsymbol{\theta}} \omega_\lambda(y)\|_\infty \leqslant \frac{1}{4\lambda} \|\nabla_{\boldsymbol{\theta}} \log p_{\boldsymbol{\theta}}(y)\|_\infty$ (Proposition A.1 of Appendix A.3). Thus, if the score $\nabla_{\boldsymbol{\theta}} \log p_{\boldsymbol{\theta}}$ is bounded, the gradient will be bounded whenever $\lambda > 0$. This property is advisable for gradient optimization and it is not guaranteed, for example, for vanilla IS ($\lambda = 0$). Thus, we can also interpret $\lambda$ as a regularization parameter for the gradient magnitude.

## 6  Data-driven Tuning of $\lambda$

The computation of the parameter $\lambda_2^*$ requires the knowledge of the divergence $I_2(P\|Q)$. Even when $P$ and $Q$ are known, computing the $I_2(P\|Q)$ can be challenging, especially for continuous

---

[6]We introduce our concentration inequalities as a one-sided bounds just for simplicity but they actually hold in both directions. Indeed, by replacing function $f$ with function $-f$, we obtain the reversed one-sided bound.

distributions, since it involves the evaluation of a complex integral.[7] In principle, we could estimate the divergence $I_2(P\|Q)$ from samples as the empirical second moment of the vanilla importance weights $\frac{1}{n}\sum_{i\in[n]}\omega(y_i)^2$. However, although possibly well-performing in practice [40], this approach would prevent any subgaussian concentration, as the behavior of the non-corrected $\omega(y)^2$ will be surely heavy-tailed whenever $\omega(y)$ is. A general-purpose approach to avoid the divergence estimation is the *Lepski's adaptation method* [32], which only requires knowing an upper and a lower bound on $I_2(P\|Q)$. Unfortunately, this method is known to be computationally intensive.

In this section, we follow a different path inspired by the recent work [66]. If a choice of the parameter $\lambda$ corrects the weight $\omega_\lambda$ leading to an ideal estimator $\widehat{\mu}_{n,\lambda}$, for the mean $\mu$, we may expect that the empirical second moment of the corrected weights $\omega_\lambda$ will provide a reasonable estimation of $I_2(P\|Q)$. Based on this observation, we propose to choose $\lambda$ by solving the following equation:

$$\lambda^2 \underbrace{\frac{1}{n}\sum_{i\in[n]}\omega_{\lambda n^{1/4}}(y_i)^2}_{\text{empirical second moment}} = \frac{2\log\frac{1}{\delta}}{3n}. \tag{3}$$

The intuition behind this approach can be stated as follows. If the empirical second moment is close to the divergence, i.e., $\frac{1}{n}\sum_{i\in[n]}\omega_{\lambda n^{1/4}}(y_i)^2 \simeq I_2(P\|Q)$, the solution $\widehat{\lambda}$ of Equation (3) approaches the optimal parameter, i.e., $\widehat{\lambda}\simeq\sqrt{(2\log(1/\delta))/(3I_2(P\|Q)n)}=\lambda_2^*$. We formalize this reasoning in Appendix A.4, proving that Equation (3) admits a unique root $\widehat{\lambda}\in[0,1]$ (Lemma A.4) and that when the number of samples $n$ grows to infinity, $\widehat{\lambda}$ converges indeed to $\lambda_2^*$ (Lemma A.8). The following result provides the concentration properties of the estimator $\widehat{\mu}_{n,\lambda}$ when using $\widehat{\lambda}$ instead of $\lambda_2^*$, under slightly more demanding requirements on the moments of the importance weights.

**Theorem 6.1.** *Let $P,Q\in\mathscr{P}(\mathcal{Y})$ be two probability distributions such that $P\ll Q$. Let $\widehat{\lambda}$ be the solution of Equation (3), then, if $I_3(P\|Q)$ is finite, for sufficiently large $n$, for every $\delta\in(0,1)$, with probability at least $1-2\delta$ it holds that:*

$$\widehat{\mu}_{n,\widehat{\lambda}} - \mu \leqslant \|f\|_\infty \frac{5+2\sqrt{3}}{2}\sqrt{\frac{2I_2(P\|Q)\log\frac{1}{\delta}}{3n}}.$$

Compared to Theorem 5.1, this result is weakened in two aspects. First, the inequality holds with a smaller probability $1-2\delta$ since two estimation processes with the same samples are needed, i.e., the computation of $\widehat{\lambda}$ and the corrected estimator $\widehat{\mu}_{n,\widehat{\lambda}}$. Second, and most important, the result holds for large $n$, whose minimum value is reported in the proof and depends on $I_3(P\|Q)$, which must be finite. We think this is a not too strong requirement considering that even the variance of an empirical estimate of $I_2(P\|Q)$ would depend on the fourth moment of the importance weight, i.e., $I_4(P\|Q)$.[8]

## 7   Related Works

Importance Sampling has a long history in Monte Carlo simulation as an effective technique for variance reduction in presence of rare events and for what-if analysis [25, 55, 20, 9, 52]. Apart from sparse exceptions [e.g., 8, 18], in the machine learning community, IS is primarily employed for off-policy estimation and learning [e.g., 11, 38, 61]. In this setting, it is well-known that IS might display an inconvenient behavior, depending on the behavioral $Q$ and target $P$ distributions [65, 40]. In particular, IS tends to enlarge the range of the estimator up to $\operatorname{ess\,sup}_{y\sim Q}p(y)/q(y)$. Although this term is finite for discrete distributions (if $P\ll Q$), it is likely unbounded for continuous ones [11]. Furthermore, in the latter case, the vanilla IS estimator might have infinite variance and tends to exhibit a heavy-tailed behavior [40, 41]. These properties suggest that a way of addressing this phenomenon is to resort to *robust* statistics, typically employed for mean estimation under heavy-tailed distributions [37]. Methods in this class include the *trimmed mean* [62, 22], the *median of means* [45, 24], and the *Catoni's* estimator [7]. For all of them, subgaussian guarantees were

---

[7]For some common distributions, including Gaussians, the integral can be computed in closed form [15].

[8]It is possible to circumvent the computation of $I_2(P\|Q)$ by choosing $\lambda$ independently from $I_2(P\|Q)$ at the price of downgrading the concentration from subgaussian to exponential (Corollary A.2 of Appendix A.4).

| Estimator | Maximum | Variance | Bias | Correction (order $\mathcal{O}$) | Concentration (order $\mathcal{O}$) | Is subgaussian? | Is unbiased when $P=Q$? | Is differentiable? |
|---|---|---|---|---|---|---|---|---|
| IS [46, 40] | $\operatorname{ess\,sup}\frac{p}{q}$ | $\frac{I_2(P\|Q)}{n}$ | $0$ | - | $\sqrt{\frac{I_2(P\|Q)}{\delta n}}$ | ✗ (poly) | ✓ | ✓ |
| SN-IS [46, 29] | $1$ | $V^{\mathrm{SN}}$ | $B^{\mathrm{SN}}$ | - | $B^{\mathrm{SN}}+\sqrt{V^{\mathrm{ES}}\log\frac{1}{\delta}}$ | ✗ (exp) | ✓ | ✓ |
| IS-TR($M$) [23, 47] | $M$ | $\frac{I_2(P\|Q)}{n}$ | $\frac{I_2(P\|Q)}{M}$ | $\sqrt{\frac{nI_2(P\|Q)}{\log\frac{1}{\delta}}}$ | $\sqrt{\frac{I_2(P\|Q)\log\frac{1}{\delta}}{n}}$ | ✓ | ✗ | ✗ |
| IS-OS($\tau$) [56] | $\frac{\sqrt{\tau}}{2}$ | $\frac{I_2(P\|Q)}{n}$ | $\frac{I_3(P\|Q)}{\tau}$ | $\sqrt[3]{\left(\frac{nI_3(P\|Q)}{\log\frac{1}{\delta}}\right)^2}$ | $\max_{\beta\in\{2,3\}}\sqrt[\beta]{\frac{I_\beta(P\|Q)(\log\frac{1}{\delta})^{\beta-1}}{n^{\beta-1}}}$ | ✗ (exp) | ✗ | ✓ |
| IS-$\lambda$ (ours) | $\frac{1}{\lambda}$ | $\frac{I_2(P\|Q)}{n}$ | $\lambda I_2(P\|Q)$ | $\sqrt{\frac{\log\frac{1}{\delta}}{I_2(P\|Q)n}}$ | $\sqrt{\frac{I_2(P\|Q)\log\frac{1}{\delta}}{n}}$ | ✓ | ✓ | ✓ |

Table 2: Comparison between several IS estimators assuming $\|f\|_\infty = 1$ and for $\alpha = 2$ w.r.t. several indexes. For the SN-IS estimator $V^{\mathrm{SN}}$ is the Efron-Stein estimate of the variance and $B^{\mathrm{SN}}$ is the bias. $V^{\mathrm{SN}}$ and $B^{\mathrm{SN}}$ converge to 0 as $n \to \infty$, but no convergence rate is provided in [29].

provided [37]. These techniques have been also successfully employed for regret minimization in finite [5] and continuous arm spaces [36]. In principle, these methods could be employed *as-is* in combination with IS, but, being general-purpose, they might disregard the peculiarities of the setting.

Several *ad-hoc* methods to cope with the problematic IS behavior have been progressively developed. An example, devised by the statistical community, is *self-normalization* [SN-IS, 46]: $\widetilde{\omega}(y_i) = \omega(y_i)/\sum_{j\in[n]}\omega(y_j)$. This approach has the advantage of controlling the range of the estimator at the price of making all samples interdependent and generating a bias. Although the asymptotic consistency is guaranteed [19, 59], its finite-sample analysis is more challenging. In [40], a polynomial concentration inequality was provided and, more recently, exponential bounds based on Efron-Stein inequalities have been proposed [28, 29]. Nevertheless, the resulting inequality is not guaranteed to decrease with $\mathcal{O}(1/\sqrt{n})$ for general distributions and it is difficult to formally relate its concentration rate to the tail properties of the involved distributions [29]. Another popular technique is the weight *truncation* (or *clipping*) [IS-TR, 23, 3]: $\omega_M^{\mathrm{TR}}(y) = \min\{\omega(y), M\}$, where $M > 0$ is the threshold. Some works rely on empirical selections of the truncation threshold [31, 10], while others focus on more theoretically principled approaches [2, 64, 47]. In particular, in [47] a subgaussian deviation bound is derived by suitably adapting the truncation threshold as a function of the number of samples $n$ and the confidence parameter $\delta$. Another interesting approach, designed for CMABs is the *switch* estimator [DR-SW, 64] that selects between DM and IS (or DR), based on the importance weight value, with also guarantees in MSE. Finally, a not so large part of the literature focuses on less crude transformations than truncation, called *smoothing* [63]. They typically take into explicit consideration the estimator tails [48], also providing asymptotic guarantees. Very recently, shrinkage transformations of the weight were proposed, based on the minimization of different bounds on the MSE, in the CMABs [56] setting. Specifically, the *optimistic shrinkage* [IS-OS, 56] leads to a transformation similar to ours $\omega_\tau^{\mathrm{OS}}(y) = \tau\omega(y)/(\omega(y)^2 + \tau)$. Unfortunately, even when knowing $P$ and $Q$ and setting $\tau$ adaptively, IS-OS is unable to achieve subgaussian concentration and requires $I_3(P\|Q)$ to be finite (Appendix E for details). Refer to Table 2 for a comparison of the estimators.

## 8 Numerical Simulations

In this section, we provide numerical simulations for off-policy evaluation (Section 8.1) and learning (Section 8.2), with the goal of showing that our estimators, while enjoying desirable theoretical properties, are competitive with traditional (e.g., vanilla IS and self-normalization) and modern baselines (e.g., truncation, optimistic shrinkage). The complete results can be found in Appendix B.[9]

### 8.1 Off-Policy Evaluation

We present two off-policy evaluation experiments. We start with a synthetic example with Gaussian distributions and, then, we move to the CMAB setting.

---

[9]The code is provided at `https://github.com/albertometelli/subgaussian-is`.

| Estimator / $n$ | 10 | 20 | 50 | 100 | 200 | 500 | 1000 |
|---|---|---|---|---|---|---|---|
| IS | $27.43 \pm 13.33$ | $15.70 \pm 4.83$ | $10.89 \pm 1.81$ | $9.26 \pm 0.92$ | $12.41 \pm 1.88$ | $9.42 \pm 0.68$ | $5.84 \pm 0.27$ |
| SN-IS | $23.89 \pm 5.77$ | $15.62 \pm 2.62$ | $10.96 \pm 1.18$ | $9.53 \pm 0.74$ | $8.82 \pm 0.62$ | $7.48 \pm 0.37$ | $5.14 \pm 0.20$ |
| IS-TR | $23.47 \pm 7.52$ | $14.03 \pm 2.75$ | $10.32 \pm 1.47$ | $8.89 \pm 0.79$ | $7.68 \pm 0.46$ | $6.21 \pm 0.28$ | $4.22 \pm 0.15$ |
| IS-OS | $19.25 \pm 8.68$ | $10.93 \pm 3.29$ | $8.37 \pm 1.35$ | $7.06 \pm 0.61$ | $8.69 \pm 1.44$ | $6.65 \pm 0.47$ | $3.97 \pm 0.16$ |
| IS-$\lambda$* | $\mathbf{21.75 \pm 6.36}$ | $\mathbf{13.17 \pm 2.45}$ | $\mathbf{9.26 \pm 1.19}$ | $7.76 \pm 0.62$ | $6.53 \pm 0.38$ | $5.29 \pm 0.23$ | $3.52 \pm 0.12$ |
| IS-$\lambda$** | $\mathbf{20.66 \pm 4.08}$ | $\mathbf{12.62 \pm 2.19}$ | $\mathbf{8.86 \pm 1.08}$ | $\mathbf{7.39 \pm 0.57}$ | $5.94 \pm 0.32$ | $4.74 \pm 0.20$ | $3.19 \pm 0.10$ |
| IS-$\widehat{\lambda}$ | $\mathbf{18.19 \pm 3.93}$ | $\mathbf{10.27 \pm 1.64}$ | $\mathbf{7.03 \pm 0.75}$ | $\mathbf{5.79 \pm 0.38}$ | $\mathbf{3.85 \pm 0.21}$ | $\mathbf{2.90 \pm 0.10}$ | $\mathbf{2.06 \pm 0.05}$ |

Table 3: Absolute error in the illustrative example varying the number of samples $n$ for the different estimators (mean $\pm$ std, 60 runs). For each column, the estimator with smallest absolute error and the ones not statistically significantly different from that one (Welch's t-test with $p < 0.02$) are in bold.

| $s$ / $\lambda$ | 0 | 0.1 | 0.2 | 0.5 |
|---|---|---|---|---|
| $-\infty$ | $3.12 \pm 0.29$ | $3.12 \pm 0.29$ | $3.12 \pm 0.29$ | $3.12 \pm 0.29$ |
| $-5$ | $6.73 \pm 1.21$ | $\mathbf{2.70 \pm 0.30}$ | $2.77 \pm 0.30$ | $2.57 \pm 0.31$ |
| $-2$ | $6.73 \pm 1.21$ | $\mathbf{2.45 \pm 0.34}$ | $\mathbf{2.42 \pm 0.32}$ | $\mathbf{2.28 \pm 0.32}$ |
| $-1$ | $6.73 \pm 1.21$ | $\mathbf{2.72 \pm 0.47}$ | $\mathbf{2.47 \pm 0.37}$ | $\mathbf{2.18 \pm 0.32}$ |
| $-0.5$ | $6.73 \pm 1.21$ | $3.44 \pm 0.64$ | $\mathbf{2.71 \pm 0.47}$ | $\mathbf{2.20 \pm 0.34}$ |
| $0$ | $6.73 \pm 1.21$ | $4.83 \pm 0.89$ | $3.66 \pm 0.68$ | $\mathbf{2.38 \pm 0.38}$ |
| $0.5$ | $6.73 \pm 1.21$ | $5.69 \pm 1.05$ | $4.85 \pm 0.89$ | $3.03 \pm 0.52$ |

Table 4: Absolute error in the illustrative example varying the parameter $s$ of the corrected weight when $n = 500$ (mean $\pm$ std, 60 runs). The estimator with smallest absolute error and the ones not statistically significantly different from that one (Welch's t-test with $p < 0.02$) are in bold.

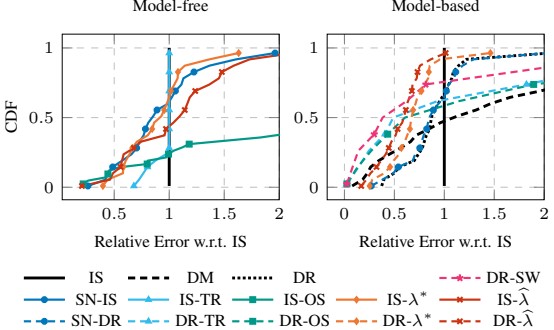

Figure 2: CDF of the absolute error normalized by IS error for stochastic rewards with noise 0.25, across 110 conditions.

### 8.1.1 Synthetic Experiment

In this experiment, we compare our corrected estimators with IS baselines in a *continuous-distribution* off-policy estimation problem. Specifically, we consider a Gaussian behavioral policy $Q = \mathcal{N}(\mu_Q, \sigma_Q^2)$ and a Gaussian target policy $P = \mathcal{N}(\mu_P, \sigma_P^2)$. We generate $n$ i.i.d. samples from $Q$ and we estimate the expectation of function $f(y) = 100 \cos(2\pi y)$ under $P$. We select $\mu_Q = 0$, $\mu_P = 0.5$, $\sigma_Q^2 = 1$ and $\sigma_P^2 = 1.9$, leading to a divergence $I_2(P \| Q) \simeq 27.9$. The results with different choices of the $\sigma_P^2$ are reported in Appendix B.1.1.

**Estimators Comparison** In Table 3, we report the absolute error between the estimated and the true mean for the different importance sampling estimators. For our correction, we report the results obtained with optimal value of $\lambda$ according to Theorem 5.1 with $\alpha = 2$ (IS-$\lambda$*), a value of $\lambda$ that optimizes a tighter bound reported in Appendix A.3 (IS-$\lambda$**), and the value estimated from samples as in Section 6 (IS-$\widehat{\lambda}$). We compare these estimators with vanilla IS (IS), self-normalized IS (SN-IS), weight truncation (IS-TR) with optimal threshold selected as in [47], and IS with optimistic shrinkage (IS-OS), where $\tau$ is computed by minimizing an MSE bound as in [56]. We notice that our estimators consistently outperform the traditional ones (IS and SN-IS) and overall suffer smaller errors than IS-TR and IS-OS. Interestingly, the minimum error is often obtained by IS-$\widehat{\lambda}$, which uses an estimated value $\widehat{\lambda}$ that tends to get a higher value than both $\lambda$* and $\lambda$**. In this way, the correction is more intense, which, in this specific example, turns out to be beneficial.

**Comparison of Different Values of** $s$ We empirically test different values of the parameter $s$ employed in Definition 4.1, in the same setting of Table 3 with $n = 500$ for the estimator IS-$\lambda$. Since for general value of $s$, we do not have a principled way to select the correction parameter $\lambda$, we consider different values of $\lambda$. The results are reported in Table 4. We can see that the best results are obtained with $s \in \{-1, -2\}$.

### 8.1.2 Contextual Bandits

In this section, we report the experiments about off-policy evaluation in CMABs.

| Estimator / $n$ | 100 | 200 | 500 | 1000 | 2000 | 5000 | 10000 | 20000 |
|---|---|---|---|---|---|---|---|---|
| IS | **17.38±1.27** | **22.26±2.00** | 15.98±0.82 | 8.36±0.21 | **4.67±0.07** | 2.68±0.03 | 2.15±0.02 | 1.10±0.00 |
| SN-IS | **23.95±1.68** | **19.39±1.30** | 17.94±0.50 | 11.43±0.22 | 7.10±0.13 | 2.54±0.03 | **1.61±0.01** | 1.10±0.00 |
| IS-TR | **17.38±1.27** | **18.92±1.36** | 15.88±0.82 | 8.36±0.21 | **4.67±0.07** | 2.68±0.03 | 2.15±0.02 | 1.10±0.00 |
| IS-OS | 24.91±1.45 | 31.93±1.15 | 15.38±0.56 | 17.25±0.45 | 16.41±0.37 | 30.63±0.15 | 33.95±0.02 | 33.61±0.01 |
| IS-λ* | **17.22±1.35** | **17.10±1.03** | 11.57±0.45 | **5.66±0.17** | 4.86±0.06 | 2.73±0.02 | 2.47±0.02 | 1.27±0.01 |
| IS-λ** | **17.21±1.39** | **16.20±0.93** | 10.93±0.37 | **5.55±0.15** | 5.05±0.06 | 2.82±0.02 | 2.68±0.02 | 1.43±0.01 |
| IS-$\widehat{\lambda}$ | **18.16±1.49** | **16.52±0.85** | 11.23±0.29 | 6.48±0.15 | 5.85±0.07 | 3.01±0.03 | 2.89±0.02 | 1.50±0.01 |
| DM | 20.52±1.18 | 25.28±0.97 | 36.19±0.31 | 36.04±0.08 | 36.95±0.06 | 41.99±0.01 | 42.70±0.01 | 42.71±0.00 |
| DR | 23.00±1.88 | 20.02±0.92 | 8.30±0.17 | **4.37±0.08** | 2.16±0.02 | **1.38±0.01** | 0.64±0.00 | |
| SN-DR | 20.89±1.45 | 23.38±1.91 | 20.79±0.74 | 10.99±0.17 | 6.48±0.11 | 2.54±0.02 | **1.52±0.01** | 0.99±0.00 |
| DR-TR | 18.48±1.13 | 15.96±0.72 | 18.58±0.23 | 15.52±0.09 | 15.45±0.07 | 20.33±0.01 | 21.05±0.01 | 20.78±0.00 |
| DR-OS | 18.47±1.17 | 18.84±0.60 | 17.10±0.39 | 12.19±0.22 | 8.86±0.11 | 17.52±0.06 | 18.40±0.02 | 19.04±0.02 |
| DR-SW | 22.83±1.25 | 16.81±1.14 | 4.59±0.18 | 4.70±0.09 | 4.86±0.06 | **0.77±0.01** | **1.38±0.01** | 0.78±0.00 |
| DR-λ* | 20.03±1.25 | 18.70±1.33 | 13.04±0.61 | 6.22±0.13 | 3.82±0.07 | 1.79±0.02 | **1.37±0.01** | **0.61±0.00** |
| DR-λ** | 19.40±1.22 | 17.29±1.17 | 11.39±0.53 | 5.60±0.12 | 3.65±0.06 | 1.67±0.02 | **1.38±0.01** | **0.63±0.00** |
| DR-$\widehat{\lambda}$ | 18.53±1.21 | 14.92±0.98 | 9.18±0.44 | 4.91±0.10 | 3.39±0.06 | 1.61±0.02 | **1.40±0.01** | **0.65±0.00** |

Table 5: Absolute error (multiplied by 100) in the *letter* dataset varying the number of samples $n$ for the different estimators, when $\alpha_b = 0.5$ and $\alpha_e = 0.9$ (mean ± std, 10 runs). For each column, the estimator with smallest absolute error and the ones not statistically significantly different from that one (Welch's t-test with $p < 0.05$) are in bold.

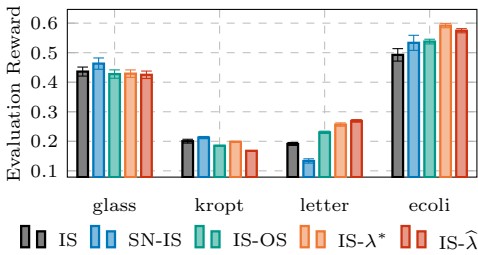

Figure 3: Evaluation reward for four different datasets after 1000 iterations (4000 iterations for *letter*) of gradient ascent with a Boltzmann policy for the model-free estimators (mean ± std, 10 runs).

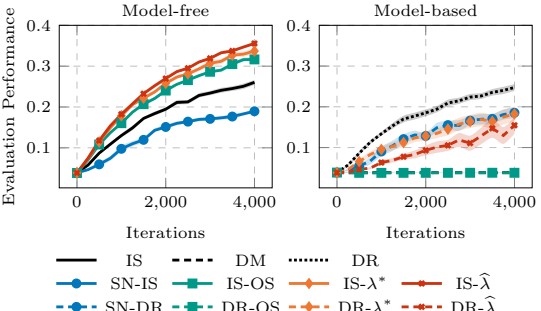

Figure 4: Evaluation reward for the *letter* dataset comparing the learning curve of different estimators (mean ± std, 10 runs).

**Setting** We follow the well-established setting of [14, 64, 57, 56]. We consider 11 UCI [13] multi-class classification datasets (see Table 9 in Appendix B.1.2). Each dataset $\mathcal{D}^* = \{(x_i, a_i^*)\}_{i \in [n^*]}$ is mapped to a CMAB problem with action set $\mathcal{A} = [K]$. Every sample $(x_i, a)$ leads to a reward given by $\mathbb{1}\{a = a_i^*\}$. To model noise, the reward is switched with probability $\nu \in [0, 1]$. Each dataset is split into a training set $\mathcal{D}_{\text{train}}$ and an evaluation $\mathcal{D}_{\text{eval}}$ with proportions 30% and 70%. A multi-class classifier C is trained on $\mathcal{D}_{\text{train}}$. The behavioral policy is obtained as: $\pi_b(a|x) = \alpha_b + \frac{1-\alpha_b}{K}$ if $a = \texttt{C}(x)$ and $\pi_b(a|x) = \frac{1-\alpha_b}{K}$ otherwise, where $\alpha_b \in [0, 1]$. The target policy $\pi_e$ is obtained as the behavioral one by training another classifier on $\mathcal{D}_{\text{train}}$ and using $\alpha_e \in [0, 1]$. We employ $\pi_b$ to generate a dataset $\mathcal{D} = \{(x_i, a_i, r_i)\}_{i \in [n]}$ sampling $x_i$ from $\mathcal{D}_{\text{eval}}$ where $a_i \sim \pi_b(\cdot|x_i)$ and $r_i$ is computed as described before. The ground truth value function is computed as $v(\pi_e) = \frac{1}{n} \sum_{x \in \mathcal{D}_{\text{eval}}} \sum_{a \in \mathcal{A}} \pi_e(a|x) r(x, a)$. For DM and DR, we employ a regressor to learn the reward with a cross-fitting procedure on the full $\mathcal{D}$.

**Estimators Comparison** We consider several settings that vary the values of $\alpha_b$ and $\alpha_e$ across all the 11 datasets, generating 110 combinations and a reward noise of $\nu = 0.25$. Details and results for the noiseless case are reported in Appendix B.1.2. To summarize the results, following the approach of [64], we plot in Figure 2 the cumulative distribution function (CDF) of the absolute error normalized by the error of IS. A lower error corresponds to a CDF curve towards the upper-left corner. We distinguish between the approaches that do not make use of the reward estimate $\widehat{r}$ (model-free, left) and the ones that do (model-based, right). As for the model-free ones, we note that the performance of our estimator IS-λ* is very close to that of SN-IS. This is likely because we are dealing with discrete distributions (actions are finite), which implicitly mitigate the degeneracy of the importance weight. Differently, the advantage w.r.t. the optimistic shrinkage (IS-OS) is quite significant. Instead, for the model-based estimators, we observe that our weight correction combined with the DR estimator (DR-λ* and DR-$\widehat{\lambda}$) outperforms the standard DR and its combinations with SN (SN-DR), truncation (DR-TR), and optimistic shrinkage (DR-OS). Instead, the switch estimator (DR-SW) displays a performance comparable to ours.

| Estimator / $\xi$ | 10 | 20 | 50 | 100 | 200 | 500 | 1000 |
|---|---|---|---|---|---|---|---|
| IS | 0.6742 | 0.5414 | 0.1754 | 0.0326 | 0.0326 | 0.0198 | 0.0014 |
| IS-$\lambda$* | 0.686 | 0.5416 | 0.176 | 0.0228 | 0.056 | 0 | 0 |
| DR | 0.6522 | 0.4094 | 0.117 | 0.0484 | 0.0378 | 0.0218 | 0.0022 |
| DR-$\lambda$* | 0.65 | 0.4046 | 0.1088 | 0.03262 | 0.009 | 0 | 0 |

Table 6: Complementary cumulative distribution of the absolute error (multiplied by 100) $\mathbb{P}(E > \xi)$ in the *glass* dataset varying the number of samples $n$ for the different estimators, when $\alpha_b = 0.9$ and $\alpha_e = 0.9999$ (5000 runs).

For the specific case of the *letter* dataset, we report in Table 5 the results obtained by setting $\alpha_b = 0.5$ and $\alpha_e = 0.9$ for different number of samples $n$. We notice essentially two behaviors. When the number of samples is very low (e.g., 100, 200) all estimators perform similarly, with poor performance. As $n$ increases, the benefits of the DR-like estimators becomes more visible. In particular, the DR-SW and our corrected estimators (DR-$\lambda$*, DR-$\lambda$**, and DR-$\widehat{\lambda}$) overall dominate the other baselines.

**Tail Behavior Experiment** We run 5000 estimation processes using the *glass* dataset, $n = 30$, $\alpha_e = 0.9999$, and $\alpha_b = 0.9$. To compare the tail behavior between vanilla weights and our correction (for both model-free and model-based estimators), we consider the absolute error random (multiplied by 100) variable $E$ (as in Table 5) and we estimate the *complementary cumulative distribution* $\mathbb{P}(E > \xi)$. Thus, for large values of $\xi$, the larger $\mathbb{P}(E > \xi)$, the heavier the tail, since a larger amount of probability mass accumulates on the right of $\xi$. Table 6 reports the results for both model-based and model-free estimators. We observe that our corrected estimators consistently display a significantly lighter tail compared to the vanilla ones.

## 8.2 Off-Policy Learning

Finally, we provide an experiment in which we employ the off-policy methods to improve a baseline policy in the CMAB framework. We refer to the same setting of Section 8.1 with a uniform behavioral policy ($\alpha_b = 0$). For the target policy, we consider a Boltzmann policy in some featurization of the context $\pi_{\boldsymbol{\theta}}(a|x) \propto \exp\left(\boldsymbol{\theta}_a^T \boldsymbol{\phi}(x)\right)$. We optimize the estimated value function in the parameters $\boldsymbol{\theta}$ via gradient ascent. Further details and experiments with regularized objectives are reported in Appendix B.2. We perform Off-PL on four datasets and the results for the model-free estimators are reported in Figure 3. We observe that our weight corrections (DR-$\lambda$* and DR-$\widehat{\lambda}$) outperform the considered baselines (IS, SN-IS, and IS-OS) on *ecoli* and *letter* datasets, whereas SN-IS emerges in the *glass* and *kropt* datasets. For the *letter* dataset, we report in Figure 4 the learning curve, distinguishing between model-free (left) and model-based (right) estimators.[10] For the model-free ones, we observe the dominance of our estimators over SN estimator (SN-IS), while the optimistic shrinkage estimator (IS-OS) behaving similarly to ours. Interestingly, for the model-based estimators, plain DR beats the other estimators, including self-normalization that performs almost identically with our DR-$\lambda$*, and IS-OS that fails completely to learn the task.

## 9 Discussion and Conclusions

In this paper, we have deepened the study of the importance sampling technique for off-policy evaluation and learning. We derived an anticoncentration bound for the vanilla IS estimator, proving polynomial concentration is tight for this setting. Then, we introduced and analyzed a class of importance weight corrections based on the intuition of smoothly shrinking the weight towards one. Assuming that the second moment of the importance weight exists, we have introduced the first transformation that achieves subgaussian concentration and maintains the differentiability of the estimator in the target policy parameters. The experimental evaluation has shown that our theoretically-grounded transformation is competitive with the traditional and modern IS baselines (including self-normalization, truncation, and optimistic shrinkage) in the CMAB framework for both evaluation and learning. The advantages of our correction are more visible in the case of continuous distributions, where the degeneracy of importance sampling is amplified. Future works include the extension of these corrections to the more challenging RL setting with continuous actions.

---

[10]Clearly, the truncated (IS-TR) and the switch (DR-SW) estimators cannot be directly employed in this setting, being non-differentiable.

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
