# Appendix

# A  Proofs and Derivations

In this section, we report the proofs of the results that are reported in the main paper.

## A.1  Proofs of Section 3

**Theorem A.1.** *Let $P, Q \in \mathscr{P}(\mathcal{Y})$ be two probability distributions such that $P \ll Q$. For every $\alpha \in (1, 2]$ and $\delta \in (0, 1)$, with probability at least $1 - \delta$ it holds that:*

$$|\widehat{\mu}_n - \mu| \leqslant \|f\|_\infty \left( \frac{2^{2-\alpha} I_\alpha(P\|Q)}{\delta n^{\alpha-1}} \right)^{\frac{1}{\alpha}}.$$

*Proof.* We first derive the following inequality concerning the $\alpha$-absolute central moment of the estimator $\widehat{\mu}_n$:

$$\mathbb{E}_{y_i \sim Q}[|\widehat{\mu}_n - \mu|^\alpha] \leqslant \mathbb{E}_{y_i \sim Q}[|\widehat{\mu}_n|^\alpha] \tag{P.1}$$

$$= \frac{1}{n^\alpha} \mathbb{E}_{y_i \sim Q}\left[ \left| \sum_{i \in [n]} \omega(y_i) f(y_i) \right|^\alpha \right]$$

$$\leqslant \frac{1}{n^{\alpha-1}} 2^{2-\alpha} \mathbb{E}_{y \sim Q}[|\omega(y)f(y)|^\alpha] \tag{P.2}$$

$$\leqslant \frac{1}{n^{\alpha-1}} 2^{2-\alpha} \|f\|_\infty I_\alpha(P\|Q), \tag{P.3}$$

where line (P.1) derives from observing that $\mu$ is the expected value of $\widehat{\mu}_n$, line (P.2) follows from Equation (1.11) of [49] using as constant $W_\alpha = 2^{2-\alpha}$ of Proposition 1.8 of [49], and line (P.3) derives from the definition of $I_\alpha$. Now we can derive the concentration inequality:

$$\mathbb{P}_{y_i \sim Q}(|\widehat{\mu}_n - \mu| \geqslant \epsilon) = \mathbb{P}_{y_i \sim Q}(|\widehat{\mu}_n - \mu|^\alpha \geqslant \epsilon^\alpha)$$

$$\leqslant \frac{\mathbb{E}_{y_i \sim Q}[|\widehat{\mu}_n - \mu|^\alpha]}{\epsilon^\alpha} \tag{P.4}$$

$$\leqslant \frac{1}{n^{\alpha-1}\epsilon^\alpha} 2^{2-\alpha} \|f\|_\infty I_\alpha(P\|Q),$$

where line (P.4) derives from Markov's inequality. By setting the right hand side of the last equation equal to $\delta$, we get the result. $\square$

**Theorem 3.1.** *There exist two distributions $P, Q \in \mathscr{P}(\mathcal{Y})$ with $P \ll Q$ and a bounded measurable function $f : \mathcal{Y} \to \mathbb{R}$ such that for every $\alpha \in (1, 2]$ and $\delta \in (0, e^{-1})$ if $n \geqslant \delta e \max\left\{1, (I_\alpha(P\|Q) - 1)^{\frac{1}{\alpha-1}}\right\}$, with probability at least $\delta$ it holds that:*

$$|\widehat{\mu}_n - \mu| \geqslant \|f\|_\infty \left( \frac{I_\alpha(P\|Q) - 1}{\delta n^{\alpha-1}} \right)^{\frac{1}{\alpha}} \left(1 - \frac{e\delta}{n}\right)^{\frac{n-1}{\alpha}}.$$

*Proof.* The proof is inspired to that of Proposition 6.2 of [7]. We construct a function $f$ and two probability measures $P$ and $Q$ that fulfill the inequality. Let $a > 0$, we consider $\mathcal{Y} = \{-a, 0, a\}$ and $f(y) = y$. First of all, we observe that $a = \|f\|_\infty$. We now define the probability distributions as follows, for $p, q \in [0, 1]$:

$$P(\{-a\}) = P(\{a\}) = \frac{p}{2} \text{ and } P(\{0\}) = 1 - p,$$

$$Q(\{-a\}) = Q(\{a\}) = \frac{q}{2} \text{ and } Q(\{0\}) = 1 - q.$$

We immediately observe that $\mathbb{E}_{y \sim P}[f(y)] = \mathbb{E}_{y \sim Q}[f(y)] = 0$. We select the values $p$ and $q$ as follows, for any $\alpha \in (1, 2]$:

$$q = \left( \frac{a}{n\epsilon} \right)^\alpha \xi,$$

$$p = \left( \frac{a}{n\epsilon} \right)^{\alpha-1} \xi,$$

where $\xi > 0$ will be specified later. First of all, we note that to make these probability valid, we need to enforce:

$$p \leqslant 1 \implies n \geqslant \frac{a}{\epsilon} \xi^{\frac{1}{\alpha}}, \tag{P.5}$$

$$q \leqslant 1 \implies n \geqslant \frac{a}{\epsilon} \xi^{\frac{1}{\alpha-1}}. \tag{P.6}$$

This choice of $p$ and $q$ ensures that $a\frac{p}{q} = n\epsilon$. Let us now compute the divergence:

$$\begin{aligned}
I_\alpha(P\|Q) &= 2\left(\frac{p}{2}\right)^\alpha \left(\frac{q}{2}\right)^{1-\alpha} + (1-p)^\alpha(1-q)^{1-\alpha} \\
&= p^\alpha q^{1-\alpha} + (1-p)^\alpha(1-q)^{1-\alpha} \\
&= \xi + \left(1 - \xi\left(\frac{a}{n\epsilon}\right)^{\alpha-1}\right)^\alpha \left(1 - \xi\left(\frac{a}{n\epsilon}\right)^\alpha\right)^{1-\alpha} \leqslant \xi + 1,
\end{aligned}$$

where the last inequality is obtained by upper bounding the second addendum under the assumption that $n \geqslant \frac{a}{\epsilon}\xi^{\frac{1}{\alpha-1}}$:

$$\left(1 - \xi\left(\frac{a}{n\epsilon}\right)^{\alpha-1}\right)^\alpha \left(1 - \xi\left(\frac{a}{n\epsilon}\right)^\alpha\right)^{1-\alpha} \leqslant \left(1 - \xi\left(\frac{a}{n\epsilon}\right)^{\alpha-1}\right)^\alpha \left(1 - \xi\left(\frac{a}{n\epsilon}\right)^{\alpha-1}\right)^{1-\alpha} = 1 - \xi\left(\frac{a}{n\epsilon}\right)^{\alpha-1} \leqslant 1.$$

Thus, we select $\xi = I_\alpha(P\|Q) - 1$. Let us now consider the vanilla IS estimator $\hat{\mu}_n$, whose expectation is $\mu = 0$, and the following derivation:

$$\begin{aligned}
\mathbb{P}_{y_i \sim Q}(|\hat{\mu}_n - \mu| > \epsilon) &= \mathbb{P}_{y_i \sim Q}(\{\hat{\mu}_n - \mu < -\epsilon\} \cup \{\hat{\mu}_n - \mu > \epsilon\}) \\
&= \mathbb{P}_{y_i \sim Q}(\hat{\mu}_n - \mu < -\epsilon) + \mathbb{P}_{y_i \sim Q}(\hat{\mu}_n - \mu > \epsilon) \tag{P.7} \\
&= 2\,\mathbb{P}_{y_i \sim Q}(\hat{\mu}_n - \mu > \epsilon), \tag{P.8}
\end{aligned}$$

where line (P.7) is obtained by observing that the two events are disjoint and line (P.8) comes from the symmetry of the events. We now lower bound the probability:

$$\begin{aligned}
\mathbb{P}_{y_i \sim Q}(\hat{\mu}_n - \mu > \epsilon) &\geqslant \mathbb{P}_{y_i \sim Q}(\text{among the } n \text{ samples, one is } a \text{ and the remaining are } 0) \\
&= n\frac{q}{2}(1-q)^{n-1} \\
&= \frac{1}{2}\left(\frac{a}{\epsilon}\right)^\alpha n^{1-\alpha}\xi\left(1 - \left(\frac{a}{n\epsilon}\right)^\alpha \xi\right)^{n-1}.
\end{aligned}$$

Now, we derive a value of $\epsilon > 0$ such that the inequality holds with probability at least $\delta$. We enforce the condition:

$$\frac{1}{2}\left(\frac{a}{\epsilon}\right)^\alpha n^{1-\alpha}\xi\left(1 - \left(\frac{a}{n\epsilon}\right)^\alpha\xi\right)^{n-1} \leqslant \delta \implies \epsilon \geqslant a\left(\frac{\xi}{\delta n^{\alpha-1}}\right)^{\frac{1}{\alpha}}\left(1 - \left(\frac{a}{n\epsilon}\right)^\alpha\xi\right)^{\frac{n-1}{\alpha}}. \tag{P.9}$$

We claim that, for $\delta \in (0, e^{-1})$, any value of $\epsilon$ fulfilling condition (P.9) must be $\epsilon \leqslant \epsilon^\star$:

$$\epsilon^\star = a\left(\frac{\xi}{\delta n^{\alpha-1}}\right)^{\frac{1}{\alpha}}\left(1 - \frac{e\delta}{n}\right)^{\frac{n-1}{\alpha}}$$

Indeed, we have:

$$\begin{aligned}
a\left(\frac{\xi}{\delta n^{\alpha-1}}\right)^{\frac{1}{\alpha}}\left(1 - \left(\frac{a}{n\epsilon^\star}\right)^\alpha\xi\right)^{\frac{n-1}{\alpha}} &= a\left(\frac{\xi}{\delta n^{\alpha-1}}\right)^{\frac{1}{\alpha}}\left(1 - \left(\frac{a}{n}\right)^\alpha\left(a\left(\frac{\xi}{\delta n^{\alpha-1}}\right)^{\frac{1}{\alpha}}\left(1 - \frac{e\delta}{n}\right)^{\frac{n-1}{\alpha}}\right)^{-\alpha}\xi\right)^{\frac{n-1}{\alpha}} \\
&= a\left(\frac{\xi}{\delta n^{\alpha-1}}\right)^{\frac{1}{\alpha}}\left(1 - \frac{\delta}{n}\left(1 - \frac{e\delta}{n}\right)^{-(n-1)}\right)^{\frac{n-1}{\alpha}} \\
&\geqslant a\left(\frac{\xi}{\delta n^{\alpha-1}}\right)^{\frac{1}{\alpha}}\left(1 - \frac{\delta e}{n}\right)^{\frac{n-1}{\alpha}} = \epsilon^\star,
\end{aligned}$$

where the last inequality derives from observing that $\left(1 - \frac{e\delta}{n}\right)^{-(n-1)} \leqslant e$ if $\delta \in (0, e^{-1})$. Finally, we rephrase conditions (P.5) and (P.6):

$$n \geqslant \frac{a}{\epsilon^\star}\xi^{\frac{1}{\alpha}} \implies n \geqslant n^{1-\frac{1}{\alpha}}\delta^{\frac{1}{\alpha}}\left(1 - \frac{e\delta}{n}\right)^{-\frac{n-1}{\alpha}} \implies n \geqslant \delta e,$$

$$n \geqslant \frac{a}{\epsilon^\star}\xi^{\frac{1}{\alpha-1}} \implies n \geqslant n^{1-\frac{1}{\alpha}}\delta^{\frac{1}{\alpha}}\xi^{\frac{1}{\alpha(\alpha-1)}}\left(1 - \frac{e\delta}{n}\right)^{-\frac{n-1}{\alpha}} \implies n \geqslant \delta e\xi^{\frac{1}{\alpha-1}},$$

having observed, again, that $\left(1 - \frac{e\delta}{n}\right)^{-\frac{n-1}{\alpha}} \leqslant e^{\frac{1}{\alpha}}$. Thus, we enforce the condition $n \geqslant \delta e \max\left\{1, \xi^{\frac{1}{\alpha-1}}\right\}$. $\qquad\square$

**Corollary A.1.** *There exist two distributions $P, Q \in \mathscr{P}(\mathcal{Y})$ with $P \ll Q$ and a bounded measurable function $f : \mathcal{Y} \to \mathbb{R}$ such that for every $\alpha \in (1, 2]$ it holds that:*

$$\underset{y_i \sim Q}{\mathbb{E}}[|\widehat{\mu}_n - \mu|^\alpha] \geqslant \|f\|_\infty^\alpha \frac{I_\alpha(P\|Q) - 1}{n^{\alpha-1}}.$$

*Proof.* Let us denote the bad event:

$$\mathcal{E} = \left\{ |\widehat{\mu}_n - \mu| \geqslant \|f\|_\infty \left( \frac{I_\alpha(P\|Q) - 1}{\delta n^{\alpha-1}} \right)^{\frac{1}{\alpha}} \left( 1 - \frac{e\delta}{n} \right)^{\frac{n-1}{\alpha}} \right\}$$

From Theorem 3.1, we know that $\mathbb{P}_{y_i \sim Q}(\mathcal{E}) \geqslant \delta$. Let us consider the expected absolute error with exponent $\alpha \in (1, 2]$ and apply the law of total expectation:

$$\underset{y_i \sim Q}{\mathbb{E}}[|\widehat{\mu}_n - \mu|^\alpha] = \underset{y_i \sim Q}{\mathbb{E}}[(\widehat{\mu}_n - \mu)^\alpha \,|\, \mathcal{E}] \, \underset{y_i \sim Q}{\mathbb{P}}(\mathcal{E}) + \underset{y_i \sim Q}{\mathbb{E}}[(\widehat{\mu}_n - \mu)^\alpha \,|\, \mathcal{E}^c] \, \underset{y_i \sim Q}{\mathbb{P}}(\mathcal{E}^c)$$

$$\geqslant \|f\|_\infty^\alpha \frac{I_\alpha(P\|Q) - 1}{\delta n^{\alpha-1}} \left( 1 - \frac{e\delta}{n} \right)^{n-1} \delta + 0.$$

The result is obtained by setting $\delta \to 0$. $\qquad\qquad\qquad\qquad\qquad\qquad\qquad\qquad\qquad\quad\square$

## A.2   Proofs of Section 4

**Lemma 4.1.** *Let $P, Q \in \mathscr{P}(\mathcal{Y})$ be two probability distributions with $P \ll Q$, then for every $\lambda \in [0, 1]$ and $y \in \mathcal{Y}$ it holds that:*
   *(i)  if $s \leqslant s'$ then $\omega_{\lambda,s}(y) \leqslant \omega_{\lambda,s'}(y)$;*
   *(ii)  if $s < 0$ then $\omega_{\lambda,s}(y) \leqslant \lambda^{\frac{1}{s}}$, otherwise if $s > 0$ then $\omega_{\lambda,s}(y) \geqslant \lambda^{\frac{1}{s}}$;*
   *(iii)  if $s < 1$ then $\mathbb{E}_{y \sim Q}[\omega_{\lambda,s}(y)] \leqslant 1$, otherwise if $s > 1$ then $\mathbb{E}_{y \sim Q}[\omega_{\lambda,s}(y)] \geqslant 1$.*

*Proof.* Recall that $\omega_{s,\lambda}(y)$ is the power mean of exponent $s$ between $\omega(y)$ and 1 and weights $(1-\lambda, \lambda)$. Consequently, (i) follows from the generalized mean inequality [6]. Let us move to (ii), if $s < 0$, we have:

$$\omega_{\lambda,s}(y) = \left( (1-\lambda)\omega(y)^s + \lambda \right)^{\frac{1}{s}} = \frac{1}{\left( \frac{1-\lambda}{\omega(y)^{-s}} + \lambda \right)^{\frac{1}{-s}}} \leqslant \lambda^{\frac{1}{s}}.$$

Instead for $s > 0$, we have:

$$\omega_{\lambda,s}(y) = \left( (1-\lambda)\omega(y)^s + \lambda \right)^{\frac{1}{s}} \geqslant \lambda^{\frac{1}{s}}.$$

Concerning (iii), let us first observe that for every $\lambda \in [0,1]$ and $s = 1$, it holds that $\mathbb{E}_{y \sim Q}[\omega_{\lambda,1}(y)] = 1$. Following from (i). and from the monotonicity of the expectation, we have that for $s < 1$:

$$\omega_{\lambda,s}(y) \leqslant \omega_{\lambda,1}(y) \quad \Longrightarrow \quad \underset{y \sim Q}{\mathbb{E}}[\omega_{\lambda,s}(y)] \leqslant \underset{y \sim Q}{\mathbb{E}}[\omega_{\lambda,1}(y)] = 1.$$

Symmetrically, for $s > 1$ we have:

$$\omega_{\lambda,s}(y) \geqslant \omega_{\lambda,1}(y) \quad \Longrightarrow \quad \underset{y \sim Q}{\mathbb{E}}[\omega_{\lambda,s}(y)] \geqslant \underset{y \sim Q}{\mathbb{E}}[\omega_{\lambda,1}(y)] = 1.$$

$\qquad\qquad\qquad\qquad\qquad\qquad\qquad\qquad\qquad\qquad\qquad\qquad\qquad\qquad\qquad\qquad\quad\square$

## A.3   Proofs of Section 5

Before going to the proofs, we introduce the following integral:

$$J_\alpha(P\|Q) = \int_{\mathcal{Y}} q(y) \left| \frac{p(y)}{q(y)} - 1 \right|^\alpha \mathrm{d}y.$$

For $\alpha = 1$, $J_1(P\|Q)$ reduces to the total variation divergence. For general values of $\alpha$, $J_\alpha(P\|Q)$ represents the $\chi^\alpha$-divergence [35, 54]. $J_\alpha(P\|Q)$ can be also seen as the $\alpha$-absolute central moment of the importance weight $\omega(y) = p(y)/q(y)$. Consequently, we immediacy conclude that $J_\alpha(P\|Q) \leqslant I_\alpha(P\|Q)$. In particular, for $\alpha = 2$, we have $J_2(P\|Q) = I_2(P\|Q) - 1$.

**Lemma A.1.** *Let $P, Q \in \mathscr{P}(\mathcal{Y})$ two probability distributions with $P \ll Q$. For every $\lambda \in [0, 1]$, the $(\lambda, -1)$-corrected importance weight induces a bias that can be bounded for every $\alpha \in (1, 2]$ as:*

$$\left| \underset{y_i \sim Q}{\mathbb{E}}[\widehat{\mu}_{n,\lambda}] - \mu \right| \leqslant \|f\|_\infty \lambda^{\alpha-1} J_\alpha(P\|Q)^{\frac{1}{\alpha}} \left[ (1-\lambda) I_\alpha(P\|Q) + \lambda \right]^{1 - \frac{1}{\alpha}}.$$

*Proof.* Let us consider the following derivation:

$$\left|\mathop{\mathbb{E}}_{y_i \sim Q}[\widehat{\mu}_{n,\lambda}] - \mu\right| = \left|\mathop{\mathbb{E}}_{y_i \sim Q}[\widehat{\mu}_{n,\lambda} - \widehat{\mu}_n]\right| \leqslant = \mathop{\mathbb{E}}_{y_i \sim Q}[|\widehat{\mu}_{n,\lambda} - \widehat{\mu}_n|] \leqslant \|f\|_\infty \mathop{\mathbb{E}}_{y \sim Q}[|\omega_\lambda(y) - \omega(y)|].$$

Thus, we have for $\alpha \in (1, 2]$:

$$\begin{aligned}
\mathop{\mathbb{E}}_{y \sim Q}[|\omega_\lambda(y) - \omega(y)|] &= \mathop{\mathbb{E}}_{y \sim Q}\left[\left|\frac{\omega(y)}{1 - \lambda + \lambda\omega(y)} - \omega(y)\right|\right]\\
&= \lambda \mathop{\mathbb{E}}_{y \sim Q}\left[\frac{|\omega(y) - 1|}{\frac{1-\lambda}{\omega(y)} + \lambda}\right]\\
&= \lambda \mathop{\mathbb{E}}_{y \sim Q}\left[|\omega(y) - 1|\left(\frac{1}{\frac{1-\lambda}{\omega(y)} + \lambda}\right)^{\alpha-1}\left(\frac{1}{\frac{1-\lambda}{\omega(y)} + \lambda}\right)^{2-\alpha}\right]\\
&\leqslant \lambda \sup_{v \geqslant 0}\left(\frac{1}{\frac{1-\lambda}{v} + \lambda}\right)^{2-\alpha} \mathop{\mathbb{E}}_{y \sim Q}\left[|\omega(y) - 1|\left(\frac{1}{\frac{1-\lambda}{\omega(y)} + \lambda}\right)^{\alpha-1}\right].
\end{aligned}$$

Concerning the first term, we observe that the function $\frac{1}{\frac{1-\lambda}{v} + \lambda}$ is monotonically increasing in $v$ and, consequently:

$$\sup_{v \geqslant 0}\left(\frac{1}{\frac{1-\lambda}{v} + \lambda}\right)^{2-\alpha} = \lim_{v \to \infty}\left(\frac{1}{\frac{1-\lambda}{v} + \lambda}\right)^{2-\alpha} = \frac{1}{\lambda^{2-\alpha}}.$$

Concerning the second term, we proceed as follows:

$$\begin{aligned}
\mathop{\mathbb{E}}_{y \sim Q}\left[|\omega(y) - 1|\left(\frac{1}{\frac{1-\lambda}{\omega(y)} + \lambda}\right)^{\alpha-1}\right] &\leqslant \mathop{\mathbb{E}}_{y \sim Q}[|\omega(y) - 1|^\alpha]^{\frac{1}{\alpha}} \mathop{\mathbb{E}}_{y \sim Q}\left[\left(\frac{1}{\frac{1-\lambda}{\omega(y)} + \lambda}\right)^\alpha\right]^{1-\frac{1}{\alpha}} &&\text{(P.10)}\\
&\leqslant \mathop{\mathbb{E}}_{y \sim Q}[|\omega(y) - 1|^\alpha]^{\frac{1}{\alpha}} \mathop{\mathbb{E}}_{y \sim Q}[((1-\lambda)\omega(y) + \lambda)^\alpha]^{1-\frac{1}{\alpha}} &&\text{(P.11)}\\
&\leqslant \mathop{\mathbb{E}}_{y \sim Q}[|\omega(y) - 1|^\alpha]^{\frac{1}{\alpha}} \mathop{\mathbb{E}}_{y \sim Q}[(1-\lambda)\omega(y)^\alpha + \lambda]^{1-\frac{1}{\alpha}} &&\text{(P.12)}\\
&= J_\alpha(P\|Q)^{\frac{1}{\alpha}}[(1-\lambda)I_\alpha(P\|Q) + \lambda]^{1-\frac{1}{\alpha}},
\end{aligned}$$

where line (P.10) derives from Hölder's inequality with exponents $\alpha$ and $\frac{\alpha}{\alpha-1}$, line (P.11) is obtained from the power mean inequality [6] by bounding the harmonic mean with the arithmetic mean, line (P.12) follows from Jensen's inequality having observed that the function $\cdot^\alpha$ is a convex function. $\square$

**Lemma A.2.** *Let $P, Q \in \mathscr{P}(\mathcal{Y})$ two probability distributions with $P \ll Q$. For every $\lambda \in [0, 1]$, the $(\lambda, -1)$-corrected importance weight induces a variance that can be bounded for every $\alpha \in (1, 2]$ as:*

$$\mathop{\mathbb{V}\mathrm{ar}}_{y_i \sim Q}[\widehat{\mu}_{n,\lambda}] \leqslant \frac{\|f\|_\infty^2}{n\lambda^{2-\alpha}}[(1-\lambda)I_\alpha(P\|Q) + \lambda].$$

*Proof.* Let us consider the following derivation:

$$\mathop{\mathbb{V}\mathrm{ar}}_{y_i \sim Q}[\widehat{\mu}_{n,\lambda}] = \frac{1}{n}\mathop{\mathbb{V}\mathrm{ar}}_{y \sim Q}[\omega_\lambda(y)f(y)] \leqslant \frac{1}{n}\mathop{\mathbb{E}}_{y \sim Q}[\omega_\lambda(y)^2 f(y)^2] \leqslant \frac{1}{n}\|f\|_\infty^2 \mathop{\mathbb{E}}_{y \sim Q}[\omega_\lambda(y)^2].$$

Thus, we have for $\alpha \in (1, 2]$:

$$\begin{aligned}
\mathop{\mathbb{E}}_{y \sim Q}[\omega_\lambda(y)^2] &= \mathop{\mathbb{E}}_{y \sim Q}\left[\left(\frac{1}{\frac{1-\lambda}{\omega(y)} + \lambda}\right)^2\right]\\
&= \mathop{\mathbb{E}}_{y \sim Q}\left[\left(\frac{1}{\frac{1-\lambda}{\omega(y)} + \lambda}\right)^\alpha\left(\frac{1}{\frac{1-\lambda}{\omega(y)} + \lambda}\right)^{2-\alpha}\right]\\
&\leqslant \sup_{v \geqslant 0}\left(\frac{1}{\frac{1-\lambda}{v} + \lambda}\right)^{2-\alpha} \mathop{\mathbb{E}}_{y \sim Q}\left[\left(\frac{1}{\frac{1-\lambda}{\omega(y)} + \lambda}\right)^\alpha\right]\\
&\leqslant \frac{1}{\lambda^{2-\alpha}}[(1-\lambda)I_\alpha(P\|Q) + \lambda],
\end{aligned}$$

where the last line is obtained by employing analogous derivations as in Lemma A.1. $\square$

**Lemma 5.1.** *Let $P, Q \in \mathscr{P}(\mathcal{Y})$ be two probability distributions with $P \ll Q$. For every $\lambda \in [0,1]$, the bias and variance of the $(\lambda, -1)$-corrected importance weight can be bounded for every $\alpha \in (1,2]$ as:*

$$\left| \mathop{\mathbb{E}}_{y \sim Q} [\widehat{\mu}_{n,\lambda}] - \mu \right| \leqslant \|f\|_\infty \lambda^{\alpha-1} I_\alpha(P\|Q), \qquad \mathop{\mathbb{V}\mathrm{ar}}_{y_i \sim Q} [\widehat{\mu}_{n,\lambda}] \leqslant \frac{\|f\|_\infty^2}{n\lambda^{2-\alpha}} I_\alpha(P\|Q).$$

*Proof.* The bias result follows immediately from Lemma A.1 by recalling that $J_\alpha(P\|Q) \leqslant I_\alpha(P\|Q)$ and observing that $(1-\lambda)I_\alpha(P\|Q) + \lambda \leqslant I_\alpha(P\|Q)$ as $I_\alpha(P\|Q) \geqslant 1$. The variance result is obtained from Lemma A.2 by observing that $(1-\lambda)I_\alpha(P\|Q) + \lambda \leqslant I_\alpha(P\|Q)$ as $I_\alpha(P\|Q) \geqslant 1$. $\qquad\square$

**Lemma A.3.** *Let $P, Q \in \mathscr{P}(\mathcal{Y})$ two probability distributions such that $P \ll Q$. Let $\{y_i\}_{i \in [n]}$ sampled independently from $Q$. For every $\alpha \in (1,2]$ and $\delta \in (0,1)$ then, for every $\lambda \in [0,1]$, with probability at least $1 - \delta$ it holds that:*

$$\widehat{\mu}_{n,\lambda} - \mu \leqslant \|f\|_\infty \sqrt{\frac{2\log\frac{1}{\delta}}{n\lambda^{2-\alpha}} \left[(1-\lambda)I_\alpha(P\|Q) + \lambda\right]} + \frac{2\|f\|_\infty \log\frac{1}{\delta}}{3\lambda n}$$
$$+ \|f\|_\infty \lambda^{\alpha-1} J_\alpha(P\|Q)^{\frac{1}{\alpha}} \left[(1-\lambda)I_\alpha(P\|Q) + \lambda\right]^{1-\frac{1}{\alpha}}.$$

*Proof.* The proof is a straightforward application of Bernstein's inequality together with Lemma A.1 and Lemma A.2. First of all, we highlight the bias in the following decomposition:

$$\widehat{\mu}_{n,\lambda} - \mu = \underbrace{\widehat{\mu}_{n,\lambda} - \mathop{\mathbb{E}}_{y_i \sim Q}[\widehat{\mu}_{n,\lambda}]}_{\text{concentration}} + \underbrace{\mathop{\mathbb{E}}_{y_i \sim Q}[\widehat{\mu}_{n,\lambda}] - \mu}_{\text{bias}}.$$

The bias term is bounded by using Lemma A.1, while for the concentration term we apply Bernstein's inequality. Let $\delta \in (0,1)$, with probability at least $1 - \delta$ it holds that:

$$\widehat{\mu}_{n,\lambda} - \mathop{\mathbb{E}}_{y_i \sim Q}[\widehat{\mu}_{n,\lambda}] \leqslant \sqrt{2 \mathop{\mathbb{V}\mathrm{ar}}_{y_i \sim Q}[\widehat{\mu}_{n,\lambda}] \log\frac{1}{\delta}} + \frac{2\|\mu_\lambda\|_\infty \log\frac{1}{\delta}}{3n}$$
$$\leqslant \|f\|_\infty \sqrt{\frac{2\log\frac{1}{\delta}}{n\lambda^{2-\alpha}} \left[(1-\lambda)I_\alpha(P\|Q) + \lambda\right]} + \frac{2\|f\|_\infty \log\frac{1}{\delta}}{3\lambda n},$$

where the last line is obtained by bounding the variance with Lemma A.2 and recalling that $\|\mu_\lambda\|_\infty \leqslant \frac{\|f\|_\infty}{\lambda}$. $\qquad\square$

We discuss how to optimize this bound in $\lambda$ in Appendix C. We now move to a simplified version of the bound.

**Theorem 5.1.** *Let $P, Q \in \mathscr{P}(\mathcal{Y})$ be two probability distributions such that $P \ll Q$. For every $\alpha \in (1,2]$ and $\delta \in (0,1)$, if we select $\lambda = \lambda_\alpha^*$ then, with probability at least $1 - \delta$ it holds that:*

$$\widehat{\mu}_{n,\lambda_\alpha^*} - \mu \leqslant \|f\|_\infty (2 + \sqrt{3}) \left( \frac{2I_\alpha(P\|Q)^{\frac{1}{\alpha-1}} \log\frac{1}{\delta}}{3(\alpha-1)^2 n} \right)^{1-\frac{1}{\alpha}}, \quad \text{with} \quad \lambda_\alpha^* = \left( \frac{2\log\frac{1}{\delta}}{3(\alpha-1)^2 I_\alpha(P\|Q)n} \right)^{\frac{1}{\alpha}}.$$

*Proof.* The derivation is analogous to that of Lemma A.3 using Bernstein's inequality and Lemma 5.1, leading to the inequality:

$$\widehat{\mu}_{n,\lambda} - \mu \leqslant \|f\|_\infty \sqrt{\frac{2\log\frac{1}{\delta}}{n\lambda^{2-\alpha}} I_\alpha(P\|Q)} + \frac{2\log\frac{1}{\delta}}{3\lambda n}\|f\|_\infty + \|f\|_\infty \lambda^{\alpha-1} I_\alpha(P\|Q) \tag{P.13}$$

This is a convex function of $\lambda$ that can be minimized by vanishing the derivative. The derivative is actually a quadratic function in $\lambda^{\frac{\alpha}{2}}$ and its positive solution has a quite complex expression:

$$\lambda_\alpha^\# := \left( \frac{-3\alpha + \sqrt{3}\sqrt{(\alpha+2)(3\alpha-2)} + 6}{6\sqrt{2}(\alpha-1)} \right)^{\frac{2}{\alpha}} \left( \frac{\log\frac{1}{\delta}}{I_\alpha(P\|Q)n} \right)^{\frac{2}{\alpha}} \leqslant \left( \frac{2}{3(\alpha-1)^2} \right)^{\frac{1}{\alpha}} \left( \frac{\log\frac{1}{\delta}}{I_\alpha(P\|Q)n} \right)^{\frac{1}{\alpha}} =: \lambda_\alpha^*,$$

where the inequality holds with equality when $\alpha = 2$. By substituting this value of $\lambda_\alpha^*$ we obtain the bound:

$$\widehat{\mu}_{n,\lambda_\alpha^*} - \mu \leqslant \|f\|_\infty \left( 2 - \sqrt{3} + \alpha(-2 + \sqrt{3} + \alpha) \right) \left( \frac{2I_\alpha(P\|Q)^{\frac{1}{\alpha-1}} \log\frac{1}{\delta}}{3(\alpha-1)^2 n} \right)^{1-\frac{1}{\alpha}}$$

$$\leqslant \|f\|_\infty (2+\sqrt{3}) \left( \frac{2I_\alpha(P\|Q)^{\frac{1}{\alpha-1}} \log\frac{1}{\delta}}{3(\alpha-1)^2 n} \right)^{1-\frac{1}{\alpha}},$$

having observed that $\left(2-\sqrt{3}+\alpha(-2+\sqrt{3}+\alpha)\right)$ is a monotonically increasing function of $\alpha$. $\qquad\square$

**Remark A.1.** *In the proof of Theorem 5.1, we did not consider the possibility that $\lambda_\alpha^* > 1$, that would lead to a non-valid correction parameter. We claim that this circumstance occurs for very small values of $n$ and $\delta$ only. Indeed:*

$$\lambda_\alpha^* \leqslant 1 \quad\Longrightarrow\quad n \geqslant \frac{2\log\frac{1}{\delta}}{3(\alpha-1)^2 I_\alpha(P\|Q)}.$$

*In any case, if it occurs that $\lambda_\alpha^* > 1$, we conventionally clip it to 1.*

**Proposition A.1.** *Let $\lambda \in [0,1]$. For every $y \in \mathcal{Y}$, let $\omega(y) = \frac{p_\theta(y)}{q(y)}$, for a target distribution $p_\theta$ differentiable in $\theta$. Then, it holds that:*

$$\|\nabla_\theta \omega_\lambda(y)\|_\infty \leqslant \frac{1}{4\lambda} \|\nabla_\theta \log p_\theta(y)\|_\infty.$$

*Proof.* Let us first compute the gradient explicitly:

$$\nabla_\theta \omega_\lambda(y) = \frac{\partial\omega_\lambda}{\partial\omega}(y)\nabla_\theta\omega(y) = \frac{1-\lambda}{(1-\lambda+\lambda\omega(y))^2}\omega(y)\nabla_\theta\log p_\theta(y)$$

To get the result, we maximize the value of the following function:

$$g(v) = \frac{(1-\lambda)v}{(1-\lambda+\lambda v)^2}.$$

First of all, we observe that for $v=0$ and $v\to\infty$, the function has value 0. Thus, the maximum must lie in between. We vanish the derivative to find it:

$$\frac{\partial g(v)}{\partial v} = \frac{(1-\lambda)(1-\lambda-\lambda v)}{(1-\lambda+\lambda v)^3} = 0 \quad\Longrightarrow\quad v^* = \frac{1}{\lambda} - 1.$$

By substituting the found value, we obtain:

$$g(v^*) = \frac{1}{4\lambda}.$$

The result is obtained by applying the $L_\infty$-norm. $\qquad\square$

## A.4 Proofs of Section 6

For the sake of simplicity, we will denote with $\eta = \lambda n^{1/4}$. We introduce the following equation:

$$h(\eta) = \eta^2 \mathop{\mathbb{E}}_{y\sim Q}\left[\omega_\eta(y)^2\right] = \frac{2\log\frac{1}{\delta}}{3\sqrt{n}},$$

and we denote with $\eta^\dagger$ a solution of this equation. We introduce the corresponding empirical version, that is equivalent to Equation (3):

$$\widehat{h}(\eta) = \frac{\eta^2}{n}\sum_{i\in[n]}\omega_\eta(y_i)^2 = \frac{2\log\frac{1}{\delta}}{3\sqrt{n}},$$

having $\widehat{\eta}$ as solution. Clearly, we have $\mathbb{E}_{y_i\sim Q}\left[\widehat{h}(\eta)\right] = h(\eta)$.

**Lemma A.4.** *Let $h(\eta) = \eta^2 \mathbb{E}_{y\sim Q}\left[\omega_\eta(y)^2\right]$. The following properties hold:*

*(i) for every $\eta \in [0,1]$ we have $h(\eta) \in [0,1]$;*
*(ii) for every $c \in (0,1]$, the equation $h(\eta) = c$ admits at most one solution.*

*Proof.* For (i) we immediately observe that $h(\eta) \geqslant 0$. Moreover, we have $\omega_\eta(y) \leqslant \eta^{-1}$, from which the result follows. For (ii) we show that $h(\eta)$ is monotonically increasing in $\eta$:

$$\frac{\partial h}{\partial\eta}(\eta) = 2\eta \mathop{\mathbb{E}}_{y\sim Q}\left[\frac{\omega(y)^2}{(1-\eta+\eta\omega(y))^3}\right] > 0.$$

$\qquad\square$

**Remark A.2.** *It might be the case that the equation $\widehat{h}(\eta) = \frac{2\log\frac{1}{\delta}}{3\sqrt{n}}$ admits no solution, for instance when $\frac{2\log\frac{1}{\delta}}{3\sqrt{n}} > 1$ or when $\sup_{\eta\in[0,1]} \widehat{h}(\eta) < 1$. In these cases, we conventionally set the solution $\eta^\dagger = 1$. We stress that this circumstance occurs only for small values of $n$, as in Remark A.1. Indeed, the right hand side $\frac{2\log\frac{1}{\delta}}{3\sqrt{n}} \to 0$ when $n \to \infty$.*

**Lemma A.5.** *Let $h(\eta) = \eta^2\, \mathbb{E}_{y\sim Q}\left[\omega_\eta(y)^2\right]$. Let $\eta^\dagger \in [0,1]$ such that:*

$$h(\eta^\dagger) = \frac{2\log\frac{1}{\delta}}{3\sqrt{n}} \quad and \quad \lambda^\dagger = \eta^\dagger n^{-1/4}$$

*then it holds that:*

$$\lambda_2^* \leqslant \lambda^\dagger \leqslant \sqrt{2}\lambda_2^*,$$

*where the second inequality holds if $n \geqslant \frac{4096(I_3(P\|Q) - I_2(P\|Q))^4 \left(\log\frac{1}{\delta}\right)^2}{9I_2(P\|Q)^6}$, whenever $I_3(P\|Q)$ is finite.*

*Proof.* Let us first observe that:

$$\mathbb{E}_{y\sim Q}\left[\omega_\eta(y)^2\right] = \mathbb{E}_{y\sim Q}\left[\frac{1}{\left(\frac{1-\eta}{\omega(y)}+\eta\right)^2}\right] \leqslant \mathbb{E}_{y\sim Q}\left[((1-\eta)\omega(y)+\eta)^2\right] = (I_2(P\|Q)-1)\eta^2 + 1 \leqslant I_2(P\|Q),$$

where the first inequality derives from the inequality between the harmonic and arithmetic mean. From the last inequality, we have:

$$h(\eta) \leqslant \eta^2 I_2(P\|Q) \implies \eta^\dagger \geqslant \sqrt{\frac{2\log\frac{1}{\delta}}{3I_2(P\|Q)\sqrt{n}}} \implies \lambda^\dagger = \sqrt{\frac{2\log\frac{1}{\delta}}{3I_2(P\|Q)n}} = \lambda_2^*.$$

Concerning the lower bound, we proceed with a second order Taylor expansion centered in $\eta = 0$:

$$\frac{1}{\left(\frac{1-\eta}{\omega(y)}+\eta\right)^2} = \omega(y)^2 - 2\omega(y)^2(\omega(y)-1)\eta + 3(\omega(y)-1)^2\omega(y)^2\overline{\eta}^2 \geqslant \omega(y)^2 - 2\omega(y)^2(\omega(y)-1)\eta,$$

for some $\overline{\eta} \in [0,\eta]$. From which, we obtain:

$$\mathbb{E}_{y\sim Q}\left[\frac{1}{\left(\frac{1-\eta}{\omega(y)}+\eta\right)^2}\right] \geqslant \mathbb{E}_{y\sim Q}\left[\omega(y)^2 - 2\omega(y)^2(\omega(y)-1)\eta\right] = I_2(P\|Q) - 2\eta(I_3(P\|Q) - I_2(P\|Q)).$$

By moving to function $h(\eta)$, and recalling the equation $h(\eta) = \frac{2\log\frac{1}{\delta}}{3\sqrt{n}}$, we have:

$$h(\eta) = \eta^2\, \mathbb{E}_{y\sim Q}\left[\omega_\eta(y)^2\right] \geqslant \eta^2 I_2(P\|Q) - 2\eta^3(I_3(P\|Q) - I_2(P\|Q))$$

$$\implies \eta^2 I_2(P\|Q) - 2\eta^3(I_3(P\|Q) - I_2(P\|Q)) \leqslant \frac{2\log\frac{1}{\delta}}{3\sqrt{n}}.$$

We prove that for sufficiently large $n$, all solutions $\eta^\dagger$ of the previous inequality satisfy $\eta \leqslant \sqrt{\frac{4\log\frac{1}{\delta}}{3I_2(P\|Q)\sqrt{n}}}$:

$$\frac{4\log\frac{1}{\delta}}{3I_2(P\|Q)\sqrt{n}} I_2(P\|Q) - 2\left(\frac{4\log\frac{1}{\delta}}{3I_2(P\|Q)\sqrt{n}}\right)^{\frac{3}{2}}(I_3(P\|Q) - I_2(P\|Q)) > \frac{2\log\frac{1}{\delta}}{3\sqrt{n}}$$

$$\implies n \geqslant \frac{4096\left(I_3(P\|Q) - I_2(P\|Q)\right)^4 \left(\log\frac{1}{\delta}\right)^2}{9I_2(P\|Q)^6}.$$

This, implies that $\lambda^\dagger \leqslant \sqrt{\frac{4\log\frac{1}{\delta}}{3I_2(P\|Q)n}} = \sqrt{2}\lambda_2^*.$ □

**Lemma A.6.** *Let $h(\eta) = \eta^2\, \mathbb{E}_{y\sim Q}\left[\omega_\eta(y)^2\right]$, then it holds that:*

$$\frac{\partial h(\eta)}{\partial \eta^2} \geqslant I_2(P\|Q)^{-2}.$$

*Proof.* Let us first observe that:

$$\frac{\partial h(\eta)}{\partial \eta^2} = \frac{\partial h(\eta)}{\partial \eta}\frac{\partial \eta}{\partial \eta^2} = \frac{\partial h(\eta)}{\partial \eta}\frac{1}{2\eta}.$$

The first factor was already computed in the proof of Lemma A.4. We now lower bound it. Let us first prove the following auxiliary inequality:

$$1 = \mathop{\mathbb{E}}_{y\sim Q}\left[\omega(y)\right]^2 = \mathop{\mathbb{E}}_{y\sim Q}\left[\frac{\omega(y)}{1-\lambda+\lambda\omega(y)}(1-\lambda+\lambda\omega(y))\right]^2 \leqslant \mathop{\mathbb{E}}_{y\sim Q}\left[\frac{\omega(y)^2}{(1-\lambda+\lambda\omega(y))^2}\right]\mathop{\mathbb{E}}_{y\sim Q}\left[(1-\lambda+\lambda\omega(y))^2\right]$$

$$\leqslant \mathop{\mathbb{E}}_{y\sim Q}\left[\frac{\omega(y)^2}{(1-\lambda+\lambda\omega(y))^2}\right]I_2(P\|Q),$$

(P.14)

where the first inequality follows from Cauchy-Schwarz's and the second one by recalling that $\mathbb{E}_{y\sim Q}\left[(1-\lambda+\lambda\omega(y))^2\right]\leqslant I_2(P\|Q)$. Now, we proceed with Hölder's inequality with $p=\frac{3}{2}$ and $q=3$:

$$\mathop{\mathbb{E}}_{y\sim Q}\left[\frac{\omega(y)^2}{(1-\lambda+\lambda\omega(y))^2}\right] = \mathop{\mathbb{E}}_{y\sim Q}\left[\frac{\omega(y)^{\frac{4}{3}}}{(1-\lambda+\lambda\omega(y))^2}\omega(y)^{\frac{2}{3}}\right] \leqslant \mathop{\mathbb{E}}_{y\sim Q}\left[\frac{\omega(y)^2}{(1-\lambda+\lambda\omega(y))^3}\right]^{\frac{2}{3}}\mathop{\mathbb{E}}_{y\sim Q}\left[\omega(y)^2\right]^{\frac{1}{3}}$$

$$= \mathop{\mathbb{E}}_{y\sim Q}\left[\frac{\omega(y)^2}{(1-\lambda+\lambda\omega(y))^3}\right]^{\frac{2}{3}}I_2(P\|Q)^{\frac{1}{3}}.$$

(P.15)

Putting together Equation (P.14) and Equation (P.15), we have:

$$\mathop{\mathbb{E}}_{y\sim Q}\left[\frac{\omega(y)^2}{(1-\lambda+\lambda\omega(y))^3}\right] \geqslant \mathop{\mathbb{E}}_{y\sim Q}\left[\frac{\omega(y)^2}{(1-\lambda+\lambda\omega(y))^2}\right]^{\frac{3}{2}}I_2(P\|Q)^{-\frac{1}{2}} \geqslant I_2(P\|Q)^{-2}.$$

$\square$

**Lemma A.7.** *Let* $h(\eta)=\eta^2\,\mathbb{E}_{y\sim Q}[\omega_\eta(y)^2]$ *and* $\widehat{h}(\eta)=\frac{\eta^2}{n}\sum_{i\in[n]}\omega_\eta(y_i)^2$. *Then,* $n\widehat{h}(\eta)$ *is a self-bounding function. Therefore, for every* $\eta\in[0,1]$ *it holds that:*

$$\mathop{\Pr}_{y_i\sim Q}\left(\widehat{h}(\eta)-h(\eta)\geqslant\epsilon\right) \leqslant \exp\left(\frac{-\epsilon^2 n}{2\left(h(\eta)+\frac{\epsilon}{3}\right)}\right) \quad \textit{with} \quad \epsilon>0, \tag{4}$$

$$\mathop{\Pr}_{y_i\sim Q}\left(h(\eta)-\widehat{h}(\eta)\geqslant\epsilon\right) \leqslant \exp\left(\frac{-\epsilon^2 n}{2h(\eta)}\right) \quad \textit{with} \quad 0<\epsilon<h(\eta). \tag{5}$$

*Proof.* We consider the definition of self-bounding function provided in [4, Definition 1]. We denote with $n\widehat{h}^{k,z}(\eta)$ the function obtained from $n\widehat{h}(\eta)$ by replacing $\omega(y_k)$ with $z\geqslant 0$. We show that $n\widehat{h}(\eta)$ satisfies both conditions:

$$n\widehat{h}(\eta)-n\widehat{h}^{k,z}(\eta)=\eta^2\left(\omega_\eta(y_k)^2-z^2\right)\leqslant\eta^2\omega_\eta(y_k)^2\leqslant 1,$$

$$\sum_{k\in[n]}\left(n\widehat{h}(\eta)-n\widehat{h}^{k,z}(\eta)\right)^2=\sum_{k\in[n]}\left(\omega_\eta(y_k)^2-z^2\right)^2\leqslant\sum_{k\in[n]}\left(\eta^2\omega_\eta(y_k)^2\right)^2\leqslant\sum_{k\in[n]}\eta^2\omega_\eta(y_k)^2=n\widehat{h}(\eta).$$

having observed that $\eta\omega_\eta(y_k)\leqslant 1$. By applying the concentration inequalities for the self-bounding functions [4], we obtain that for every $\eta\in[0,1]$ and $\epsilon>0$ it holds that:

$$\mathop{\Pr}_{y_i\sim Q}\left(\widehat{h}(\eta)-h(\eta)\geqslant\epsilon\right)\leqslant\exp\left(\frac{-\epsilon^2 n}{2\left(h(\eta)+\frac{\epsilon}{3}\right)}\right).$$

Similarly, for every $\eta\in[0,1]$ and $0<\epsilon<h(\eta)$ it holds that:

$$\mathop{\Pr}_{y_i\sim Q}\left(h(\eta)-\widehat{h}(\eta)\geqslant\epsilon\right)\leqslant\exp\left(\frac{-\epsilon^2 n}{2h(\eta)}\right).$$

$\square$

**Lemma A.8.** *Let* $\eta^\dagger$ *be the solution of* $h(\eta^\dagger)=\frac{2\log\frac{1}{\delta}}{3\sqrt{n}}$ *and* $\widehat{\eta}$ *be the solution of* $\widehat{h}(\widehat{\eta})=\frac{2\log\frac{1}{\delta}}{3\sqrt{n}}$. *Then, for any* $\delta\in(0,1)$, *with probability at least* $1-\delta$ *it holds that:*

$$\frac{1}{2}\leqslant\frac{\widehat{\eta}}{\eta^\dagger}\leqslant\sqrt{2} \quad \textit{and} \quad \frac{1}{2}\leqslant\frac{\widehat{\lambda}}{\lambda^\dagger}\leqslant\sqrt{2},$$

*for* $n\geqslant\max\left\{544 I_2(P\|Q)^{12}\left(\frac{\log\frac{2}{\delta}}{\log\frac{1}{\delta}}\right)^2, \frac{4096(I_3(P\|Q)-I_2(P\|Q))^4\left(\log\frac{1}{\delta}\right)^2}{9 I_2(P\|Q)^6}\right\}.$

*Proof.* Let $\epsilon \in [0,1]$, consider the event $\left\{\left|\frac{\widehat{\eta}}{\eta^\dagger} - 1\right| > \epsilon\right\}$. Under the sub-event $\{\widehat{\eta} > (1+\epsilon)\eta^\dagger\}$ recalling that function $h$ and $\widehat{h}$ are increasing in $\eta$ we have:

$$\widehat{h}(\widehat{\eta}) - \widehat{h}(\eta^\dagger) \geqslant \widehat{h}((1+\epsilon)\eta^\dagger) - \widehat{h}(\eta^\dagger)$$
$$= \widehat{h}((1+\epsilon)\eta^\dagger) - \widehat{h}(\eta^\dagger) \pm h(\eta^\dagger) \pm h((1+\epsilon)\eta^\dagger)$$
$$= \widehat{h}((1+\epsilon)\eta^\dagger) - h((1+\epsilon)\eta^\dagger) + h(\eta^\dagger) - \widehat{h}(\eta^\dagger) + h((1+\epsilon)\eta^\dagger) - h(\eta^\dagger)$$
$$\geqslant \widehat{h}((1+\epsilon)\eta^\dagger) - h((1+\epsilon)\eta^\dagger) + h(\eta^\dagger) - \widehat{h}(\eta^\dagger) + 2I_2(P\|Q)^{-2}\epsilon(\eta^\dagger)^2,$$

where the last inequality follows from Lemma A.6 having applied:

$$h((1+\epsilon)\eta^\dagger) - h(\eta^\dagger) \geqslant I_2(P\|Q)^{-2}\left((1+\epsilon)^2 - 1\right)(\eta^\dagger)^2 = I_2(P\|Q)^{-2}(2+\epsilon)\epsilon(\eta^\dagger)^2 \geqslant 2I_2(P\|Q)^{-2}\epsilon(\eta^\dagger)^2.$$

Recalling that $\widehat{h}(\widehat{\eta}) = h(\eta^\dagger)$, the condition can be further simplified into $h((1+\epsilon)\eta^\dagger) - \widehat{h}((1+\epsilon)\eta^\dagger) \geqslant 2I_2(P\|Q)^{-2}\epsilon(\eta^\dagger)^2$. Symmetrically, under the sub-event $\{\widehat{\eta} < (1-\epsilon)\eta^\dagger\}$ we have:

$$\widehat{h}(\widehat{\eta}) - \widehat{h}(\eta^\dagger) \leqslant \widehat{h}((1-\epsilon)\eta^\dagger) - \widehat{h}(\eta^\dagger)$$
$$= \widehat{h}((1-\epsilon)\eta^\dagger) - \widehat{h}(\eta^\dagger) \pm h(\eta^\dagger) \pm h((1-\epsilon)\eta^\dagger)$$
$$= \widehat{h}((1-\epsilon)\eta^\dagger) - h((1-\epsilon)\eta^\dagger) + h(\eta^\dagger) - \widehat{h}(\eta^\dagger) + h((1-\epsilon)\eta^\dagger) - h(\eta^\dagger)$$
$$\leqslant \widehat{h}((1-\epsilon)\eta^\dagger) - h((1-\epsilon)\eta^\dagger) + h(\eta^\dagger) - \widehat{h}(\eta^\dagger) - I_2(P\|Q)^{-2}\left(1 - (1-\epsilon)^2\right)(\eta^\dagger)^2,$$

that can be simplified, as before, into the condition $\widehat{h}((1-\epsilon)\eta^\dagger) - h((1-\epsilon)\eta^\dagger) \geqslant I_2(P\|Q)^{-2}\epsilon(\eta^\dagger)^2$ since $1 - (1-\epsilon)^2 = \epsilon(2-\epsilon) \geqslant \epsilon$ being $\epsilon < 1$. Thus, we have:

$$\Pr_{y\sim Q}\left(\left|\frac{\widehat{\eta}}{\eta^\dagger} - 1\right| > \epsilon\right) = \Pr_{y\sim Q}\left(\widehat{\eta} > (1+\epsilon)\eta^\dagger\right) + \Pr_{y\sim Q}\left(\widehat{\eta} < (1-\epsilon)\eta^\dagger\right)$$
$$\leqslant \Pr_{y\sim Q}\left(h((1+\epsilon)\eta^\dagger) - \widehat{h}((1+\epsilon)\eta^\dagger) \geqslant 2I_2(P\|Q)^{-2}\epsilon(\eta^\dagger)^2\right)$$
$$+ \Pr_{y\sim Q}\left(\widehat{h}((1-\epsilon)\eta^\dagger) - h((1-\epsilon)\eta^\dagger) \geqslant I_2(P\|Q)^{-2}\epsilon(\eta^\dagger)^2\right).$$

First of all, we observe that $h((1+\epsilon)\eta^\dagger) = (1+\epsilon)^2(\eta^\dagger)^2\,\mathbb{E}_{y\sim Q}[\omega_{(1+\epsilon)\eta^\dagger}(y)^2] \leqslant 4(\eta^\dagger)^2 I_2(P\|Q)$. Now, recalling that function $h$ is self-bounding as proved in Lemma A.7, we have by Equation (5):

$$\Pr\left(h((1+\epsilon)\eta^\dagger) - \widehat{h}((1+\epsilon)\eta^\dagger) \geqslant 2I_2(P\|Q)^{-2}\epsilon(\eta^\dagger)^2\right) \leqslant \exp\left(\frac{-4I_2(P\|Q)^{-4}\epsilon^2(\eta^\dagger)^4 n}{2h((1+\epsilon)\eta^\dagger)}\right)$$
$$\leqslant \exp\left(\frac{-4I_2(P\|Q)^{-4}\epsilon^2(\eta^\dagger)^4 n}{8(\eta^\dagger)^2 I_2(P\|Q)}\right)$$
$$= \exp\left(\frac{-\epsilon^2(\eta^\dagger)^2 n}{2I_2(P\|Q)^5}\right),$$

provided that $2I_2(P\|Q)^{-2}\epsilon(\eta^\dagger)^2 \leqslant h((1+\epsilon)\eta^\dagger)$, that is fulfilled for every $\epsilon \in [0,1]$. Indeed, recalling that $h((1+\epsilon)\eta^\dagger) = (1+\epsilon)^2(\eta^\dagger)^2\,\mathbb{E}_{y\sim Q}[\omega_{(1+\epsilon)\eta^\dagger}(y)^2] \geqslant (1+\epsilon)^2(\eta^\dagger)^2 I_2(P\|Q)^{-2}$ (from Equation (P.14)), we have that $2I_2(P\|Q)^{-2}\epsilon(\eta^\dagger)^2 \leqslant (1+\epsilon)^2(\eta^\dagger)^2 I_2(P\|Q)^{-2}$ is fulfilled for every $\epsilon \in [0,1]$. Similarly, by Equation (4) and recalling that $h((1-\epsilon)\eta^\dagger) \leqslant h(\eta^\dagger) \leqslant (\eta^\dagger)^2 I_2(P\|Q)$, we have:

$$\Pr\left(\widehat{h}((1-\epsilon)\eta^\dagger) - h((1-\epsilon)\eta^\dagger) \geqslant I_2(P\|Q)^{-2}\epsilon(\eta^\dagger)^2\right) \leqslant \exp\left(\frac{-I_2(P\|Q)^{-4}\epsilon^2(\eta^\dagger)^4 n}{2\left(h((1-\epsilon)\eta^\dagger) + \frac{1}{3}I_2(P\|Q)^{-2}\epsilon(\eta^\dagger)^2\right)}\right)$$
$$\leqslant \exp\left(\frac{-I_2(P\|Q)^{-4}\epsilon^2(\eta^\dagger)^4 n}{2(\eta^\dagger)^2 I_2(P\|Q) + \frac{2}{3}I_2(P\|Q)^{-2}\epsilon(\eta^\dagger)^2}\right)$$
$$\exp\left(\frac{-3\epsilon^2(\eta^\dagger)^2 n}{8I_2(P\|Q)^5}\right),$$

having crudely bounded $I_2(P\|Q)^{-2}\epsilon \leqslant I_2(P\|Q)$. Putting these inequalities together, we obtain:

$$\Pr\left(\left|\frac{\widehat{\eta}}{\eta^\dagger} - 1\right| > \epsilon\right) \leqslant \exp\left(\frac{-\epsilon^2(\eta^\dagger)^2 n}{2I_2(P\|Q)^5}\right) + \exp\left(\frac{-3\epsilon^2(\eta^\dagger)^2 n}{2I_2(P\|Q)^5}\right) \leqslant 2\exp\left(\frac{-3\epsilon^2(\eta^\dagger)^2 n}{8I_2(P\|Q)^5}\right),$$

leading to the inequality holding with probability at least $1 - \delta$:

$$\left|\frac{\widehat{\eta}}{\eta^\dagger} - 1\right| \leqslant \sqrt{\frac{8I_2(P\|Q)^5 \log\frac{2}{\delta}}{3n(\eta^\dagger)^2}}.$$

Under Lemma A.5, we know that $\eta^\dagger \geq \sqrt{\frac{2\log\frac{1}{\delta}}{3I_2(P\|Q)\sqrt{n}}}$. From which we have:

$$\left|\frac{\widehat{\eta}}{\eta^\dagger} - 1\right| \leq \sqrt{\frac{4I_2(P\|Q)^6\log\frac{2}{\delta}}{\sqrt{n}\log\frac{1}{\delta}}}.$$

Simple calculations allow to conclude that $\frac{1}{2} \leq \frac{\widehat{\eta}}{\eta^\dagger} \leq \sqrt{2}$ for $n \geq 544 I_2(P\|Q)^{12}\left(\frac{\log\frac{2}{\delta}}{\log\frac{1}{\delta}}\right)^2$. $\qquad\square$

**Theorem 6.1.** *Let $P,Q \in \mathscr{P}(\mathcal{Y})$ be two probability distributions such that $P \ll Q$. Let $\widehat{\lambda}$ be the solution of Equation (3), then, if $I_3(P\|Q)$ is finite, for sufficiently large $n$, for every $\delta \in (0,1)$, with probability at least $1-2\delta$ it holds that:*

$$\widehat{\mu}_{n,\widehat{\lambda}} - \mu \leq \|f\|_\infty \frac{5+2\sqrt{3}}{2}\sqrt{\frac{2I_2(P\|Q)\log\frac{1}{\delta}}{3n}}.$$

*Proof.* Let us start observing that if we substitute a value of $\lambda$ that is proportional to $\lambda_2^*$ into Equation (P.13), we are able to provide the following bound for $\beta > 0$:

$$\widehat{\mu}_{n,\beta\lambda_2^*} - \mu \leq \frac{1+\sqrt{3}\beta+\beta^2}{\beta}\sqrt{\frac{2I_2(P\|Q)\log\frac{1}{\delta}}{3n}}.$$

Now, we provide sufficient conditions so that $\frac{1}{2}\lambda_2^* \leq \widehat{\lambda} \leq 2\lambda_2^*$. First of all, we know from Lemma A.5 that for sufficiently large $n$ we have $1 \leq \frac{\lambda^\dagger}{\lambda_2^*} \leq \sqrt{2}$. Second, from Lemma A.7, we know that for sufficiently large $n$ and with probability at least $1-\delta$, we have $\frac{1}{2} \leq \frac{\widehat{\lambda}}{\lambda^\dagger} \leq \sqrt{2}$. Thus, putting together these results we enforce $\frac{1}{2}\lambda_2^* \leq \widehat{\lambda} \leq 2\lambda_2^*$. Therefore, it holds with probability at least $1-2\delta$ and sufficiently large $n$ that:

$$\widehat{\mu}_{n,\widehat{\lambda}} - \mu \leq \frac{\|f\|_\infty}{2}(5+2\sqrt{3})\sqrt{\frac{2I_2(P\|Q)\log\frac{1}{\delta}}{3n}}.$$

$\qquad\square$

**Corollary A.2.** *Let $P,Q \in \mathscr{P}(\mathcal{Y})$ two probability distributions such that $P \ll Q$. Let $\{y_i\}_{i\in[n]}$ sampled independently from $Q$. For every $\delta \in (0,1)$, let*

$$\lambda^\ddagger = \sqrt{\frac{\log\frac{1}{\delta}}{n}}$$

*then, with probability at least $1-\delta$ it holds that:*

$$\widehat{\mu}_{n,\lambda^\ddagger} - \mu \leq \|f\|_\infty\sqrt{\frac{\log\frac{1}{\delta}}{n}}\left(\frac{2}{3}+\sqrt{2I_2(P\|Q)}+I_2(P\|Q)\right).$$

*Proof.* The result is simply obtained by substituting $\lambda^\ddagger$ into Equation (P.13). $\qquad\square$

# B    Experiments

In this appendix, we report the experimental details and additional experimental results.

**Infrastructure**    The experiments have been run on a machine with two CPUs Intel(R) Xeon(R) CPU E7-8880 v4 @ 2.20GHz (22 cores, 44 thread, 55 MB cache) and 128 GB RAM.

**Code**    The code is built on top of the *Open Bandit Pipeline* [53, `https://github.com/st-tech/zr-obp`], that is licensed under the Apache 2.0 License. In the attached code, the source files that have been modified are marked with an appropriate comment at the beginning.

## B.1    Off-Policy Evaluation

### B.1.1    Synthetic Example

**Experimental Details**    To accurately estimate the expectation of function $f$ under $P$, we generate at the beginning 10M from $P$ and we estimate the expectation $\mu$ with the sample mean. For all estimators with optimal parameter (truncation threshold or $\lambda$), we employ the significance value $\delta = 0.1$.

For the optimistic shrinkage transformation (IS-OS), we compute the correction parameter $\tau^*$, by minimizing an upper bound on the MSE, derived from the one presented in the paper [56], accounting for the fact that we do not have a reward estimate (we are not considering here a DR estimator):

$$\tau^* \in \underset{\tau \geqslant 0}{\arg\min} \underbrace{\widehat{\mathbb{Var}}_{y_i \sim Q} \left[ \omega_\tau^{\mathrm{OS}}(y_i) f(y_i) \right]}_{\text{estimated variance}} + \underbrace{\frac{\|f\|_\infty^2}{n} \sum_{i \in [n]} \left( \omega_\tau^{\mathrm{OS}}(y_i) - \omega(y_i) \right)^2}_{\text{estimated bias}},$$

where:

$$\widehat{\mathbb{Var}}_{y_i \sim Q} \left[ \omega_\tau^{\mathrm{OS}}(y_i) r(y_i) \right] = \frac{1}{n} \sum_{i \in [n]} \left( \omega_\tau^{\mathrm{OS}}(y_i) f(y_i) - \widehat{\mu}_\tau^{\mathrm{OS}} \right)^2, \qquad \widehat{\mu}_\tau^{\mathrm{OS}} = \frac{1}{n} \sum_{i \in [n]} \omega_\tau^{\mathrm{OS}}(y_i) f(y_i).$$

**Complete Results**    In all experiments, we employ $\mu_P = 0.5$ and $\mu_Q = 0$. The values of $\sigma_P$ and $\sigma_Q$ for the different experiments are reported in Table 7. In Table 8 and Figure 5, we report the complete results for the different settings.

| $\sigma_Q^2$ | $\sigma_P^2$ | $I_2(P\|Q)$ |
|---|---|---|
| 1 | 1.5 | 1.904 |
| 1 | 1.9 | 27.949 |
| 1 | 1.99 | $5.104e+11$ |
| 1 | 1.999 | $8.379e+109$ |

Table 7: Variance values $\sigma_Q^2$ and $\sigma_P^2$ and divergence $I_2(P\|Q)$ for the different experiments.

$$\sigma_Q^2 = 1, \ \sigma_P^2 = 1.5$$

| Estimator / $n$ | 10 | 20 | 50 | 100 | 200 | 500 | 1000 |
|---|---|---|---|---|---|---|---|
| IS | $23.52\pm5.39$ | $\mathbf{15.39\pm3.26}$ | $\mathbf{10.06\pm1.93}$ | $8.35\pm0.73$ | $6.29\pm0.32$ | $3.93\pm0.12$ | $\mathbf{2.54\pm0.06}$ |
| SN-IS | $23.09\pm4.62$ | $\mathbf{14.37\pm2.55}$ | $\mathbf{9.15\pm1.32}$ | $8.23\pm0.63$ | $6.32\pm0.31$ | $3.96\pm0.12$ | $\mathbf{2.56\pm0.06}$ |
| IS-TR | $\mathbf{20.34\pm4.66}$ | $\mathbf{13.48\pm2.59}$ | $\mathbf{8.33\pm1.08}$ | $\mathbf{7.38\pm0.47}$ | $\mathbf{5.88\pm0.27}$ | $\mathbf{3.60\pm0.11}$ | $\mathbf{2.45\pm0.06}$ |
| IS-OS | $\mathbf{16.55\pm4.13}$ | $\mathbf{11.87\pm2.79}$ | $\mathbf{7.98\pm1.21}$ | $\mathbf{6.53\pm0.52}$ | $\mathbf{5.06\pm0.26}$ | $\mathbf{3.21\pm0.10}$ | $\mathbf{2.17\pm0.05}$ |
| IS-$\lambda$* | $\mathbf{18.86\pm4.01}$ | $\mathbf{12.20\pm2.30}$ | $\mathbf{7.44\pm0.92}$ | $\mathbf{6.53\pm0.43}$ | $\mathbf{5.14\pm0.25}$ | $\mathbf{3.25\pm0.10}$ | $\mathbf{2.20\pm0.05}$ |
| IS-$\lambda$** | $\mathbf{17.85\pm3.81}$ | $\mathbf{11.32\pm2.07}$ | $\mathbf{6.89\pm0.79}$ | $\mathbf{6.00\pm0.41}$ | $\mathbf{4.81\pm0.24}$ | $\mathbf{3.07\pm0.09}$ | $\mathbf{2.11\pm0.05}$ |
| IS-$\hat{\lambda}$ | $\mathbf{17.98\pm3.83}$ | $\mathbf{11.30\pm2.07}$ | $\mathbf{6.82\pm0.77}$ | $\mathbf{5.89\pm0.40}$ | $\mathbf{4.72\pm0.23}$ | $\mathbf{3.03\pm0.09}$ | $\mathbf{2.09\pm0.05}$ |

$$\sigma_Q^2 = 1, \ \sigma_P^2 = 1.9$$

| Estimator / $n$ | 10 | 20 | 50 | 100 | 200 | 500 | 1000 |
|---|---|---|---|---|---|---|---|
| IS | $\mathbf{27.43\pm13.33}$ | $\mathbf{15.70\pm4.83}$ | $10.89\pm1.81$ | $9.26\pm0.92$ | $12.41\pm1.88$ | $9.42\pm0.68$ | $5.84\pm0.27$ |
| SN-IS | $\mathbf{23.89\pm5.77}$ | $15.62\pm2.62$ | $10.96\pm1.18$ | $9.53\pm0.74$ | $8.82\pm0.62$ | $7.48\pm0.37$ | $5.14\pm0.20$ |
| IS-TR | $\mathbf{23.47\pm7.52}$ | $\mathbf{14.03\pm2.75}$ | $10.32\pm1.47$ | $8.89\pm0.79$ | $7.68\pm0.46$ | $6.21\pm0.28$ | $4.22\pm0.15$ |
| IS-OS | $\mathbf{19.25\pm8.68}$ | $\mathbf{10.93\pm3.29}$ | $\mathbf{8.37\pm1.35}$ | $\mathbf{7.06\pm0.61}$ | $\mathbf{8.69\pm1.44}$ | $6.65\pm0.47$ | $3.97\pm0.16$ |
| IS-$\lambda$* | $\mathbf{21.75\pm6.36}$ | $\mathbf{13.17\pm2.45}$ | $\mathbf{9.26\pm1.19}$ | $\mathbf{7.76\pm0.62}$ | $6.53\pm0.38$ | $5.29\pm0.23$ | $3.52\pm0.12$ |
| IS-$\lambda$** | $\mathbf{20.66\pm4.08}$ | $\mathbf{12.62\pm2.19}$ | $\mathbf{8.86\pm1.08}$ | $\mathbf{7.39\pm0.57}$ | $5.94\pm0.32$ | $4.74\pm0.20$ | $3.19\pm0.10$ |
| IS-$\hat{\lambda}$ | $\mathbf{18.19\pm3.93}$ | $\mathbf{10.27\pm1.64}$ | $\mathbf{7.03\pm0.75}$ | $\mathbf{5.79\pm0.38}$ | $\mathbf{3.85\pm0.21}$ | $\mathbf{2.90\pm0.10}$ | $\mathbf{2.06\pm0.05}$ |

$$\sigma_Q^2 = 1, \ \sigma_P^2 = 1.99$$

| Estimator / $n$ | 10 | 20 | 50 | 100 | 200 | 500 | 1000 |
|---|---|---|---|---|---|---|---|
| IS | $24.42\pm6.54$ | $\mathbf{25.03\pm11.38}$ | $15.72\pm3.31$ | $11.10\pm1.89$ | $8.96\pm0.74$ | $6.23\pm0.32$ | $4.77\pm0.19$ |
| SN-IS | $25.50\pm5.84$ | $20.36\pm3.36$ | $13.99\pm1.56$ | $9.58\pm1.08$ | $8.73\pm0.56$ | $6.08\pm0.27$ | $4.64\pm0.16$ |
| IS-TR | $24.42\pm6.54$ | $\mathbf{25.03\pm11.38}$ | $15.72\pm3.31$ | $11.10\pm1.89$ | $8.96\pm0.74$ | $6.23\pm0.32$ | $4.77\pm0.19$ |
| IS-OS | $\mathbf{16.39\pm4.48}$ | $\mathbf{16.89\pm6.36}$ | $\mathbf{11.20\pm1.96}$ | $\mathbf{7.66\pm1.08}$ | $6.80\pm0.48$ | $4.67\pm0.21$ | $3.62\pm0.14$ |
| IS-$\lambda$* | $24.42\pm6.54$ | $\mathbf{25.03\pm11.38}$ | $15.72\pm3.31$ | $11.10\pm1.89$ | $8.96\pm0.74$ | $6.23\pm0.32$ | $4.77\pm0.19$ |
| IS-$\lambda$** | $\mathbf{18.37\pm4.65}$ | $\mathbf{12.95\pm2.18}$ | $15.72\pm3.31$ | $11.10\pm1.89$ | $8.96\pm0.74$ | $6.23\pm0.32$ | $4.77\pm0.19$ |
| IS-$\hat{\lambda}$ | $\mathbf{16.12\pm4.19}$ | $\mathbf{12.50\pm2.04}$ | $\mathbf{7.81\pm0.77}$ | $\mathbf{5.19\pm0.41}$ | $\mathbf{4.64\pm0.24}$ | $\mathbf{2.92\pm0.11}$ | $\mathbf{2.25\pm0.05}$ |

$$\sigma_Q^2 = 1, \ \sigma_P^2 = 1.999$$

| Estimator / $n$ | 10 | 20 | 50 | 100 | 200 | 500 | 1000 |
|---|---|---|---|---|---|---|---|
| IS | $\mathbf{32.44\pm30.89}$ | $\mathbf{22.29\pm11.21}$ | $19.03\pm5.26$ | $19.39\pm4.36$ | $15.83\pm2.03$ | $9.21\pm0.50$ | $6.96\pm0.26$ |
| SN-IS | $21.06\pm5.75$ | $18.00\pm3.18$ | $14.78\pm2.10$ | $11.81\pm1.39$ | $10.66\pm0.89$ | $7.94\pm0.35$ | $6.32\pm0.20$ |
| IS-TR | $\mathbf{32.44\pm30.89}$ | $\mathbf{22.29\pm11.21}$ | $19.03\pm5.26$ | $19.39\pm4.36$ | $15.83\pm2.03$ | $9.21\pm0.50$ | $6.96\pm0.26$ |
| IS-OS | $\mathbf{21.32\pm18.62}$ | $\mathbf{15.42\pm6.75}$ | $\mathbf{12.18\pm3.00}$ | $13.50\pm3.06$ | $10.62\pm1.25$ | $6.15\pm0.33$ | $4.68\pm0.16$ |
| IS-$\lambda$* | $\mathbf{32.44\pm30.89}$ | $\mathbf{22.29\pm11.21}$ | $19.03\pm5.26$ | $19.39\pm4.36$ | $15.83\pm2.03$ | $9.21\pm0.50$ | $6.96\pm0.26$ |
| IS-$\lambda$** | $\mathbf{14.87\pm3.73}$ | $\mathbf{12.81\pm2.12}$ | $\mathbf{8.26\pm0.91}$ | $\mathbf{5.07\pm0.41}$ | $\mathbf{3.56\pm0.22}$ | $\mathbf{2.54\pm0.07}$ | $\mathbf{1.43\pm0.03}$ |
| IS-$\hat{\lambda}$ | $\mathbf{13.36\pm3.47}$ | $\mathbf{11.25\pm1.67}$ | $\mathbf{7.52\pm0.85}$ | $\mathbf{5.27\pm0.43}$ | $\mathbf{3.68\pm0.20}$ | $\mathbf{2.47\pm0.10}$ | $\mathbf{2.20\pm0.05}$ |

Table 8: Absolute error in the illustrative examples varying the number of samples $n$ for the different estimators and the different settings of Table 7 (mean $\pm$ std, 60 runs). The estimator with smallest absolute error and the ones not statistically significantly different from that one (Welch's t-test with $p < 0.02$) are in bold.

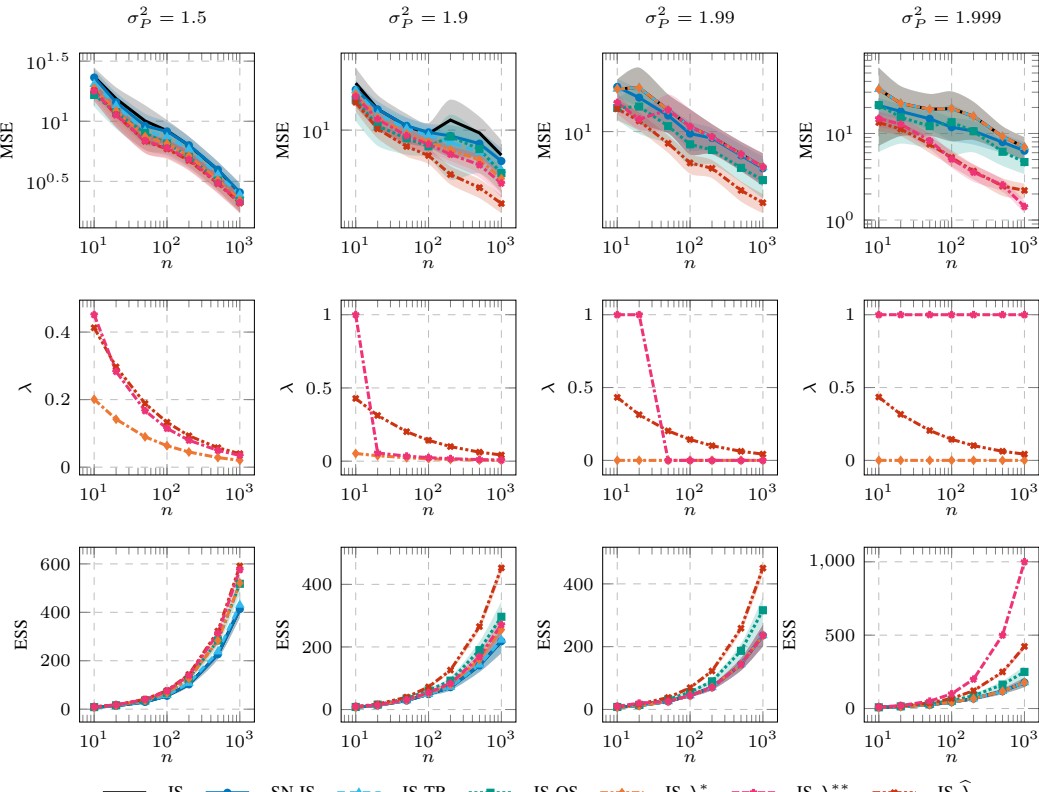

Figure 5: Mean Squared Error (MSE), correction parameter $\lambda$, and Effective Sample Size (ESS), computed as $\frac{\left(\sum_{i\in[n]}\omega(y_i)\right)^2}{\sum_{i\in[n]}\omega(y_i)^2}$, as a function of the number of samples $n$ for the different settings of Table 7 (mean $\pm$ 95% c.i., 60 runs).

### B.1.2 Contextual Bandits

**Experimental Setting** The experimental evaluation is carried out over 11 UCI Machine Learning Repository datasets [13, https://archive.ics.uci.edu/ml/index.php] as reported in Table 9. For the estimators requiring the value of the significance, we select $\delta = 0.1$.

| Dataset | ecoli | glass | isolet | kropt | letter | optdigits | page-blocks | pendigits | satimage | vehicle | yeast |
|---|---|---|---|---|---|---|---|---|---|---|---|
| Dataset size ($n^*$) | 336 | 214 | 7797 | 28056 | 20000 | 5620 | 5473 | 10992 | 6435 | 846 | 1484 |
| Context dimension | 7 | 9 | 617 | 6 | 16 | 64 | 10 | 16 | 36 | 18 | 8 |
| Classes ($K$) | 8 | 6 | 26 | 18 | 26 | 10 | 5 | 10 | 6 | 4 | 10 |

Table 9: The 11 UCI dataset considered in the experiments. For each dataset, we report the number of examples $n^*$, dimensionality of the context, and number of classes $K$.

**Complete Results** In the comprehensive experiment, we consider 110 combinations obtained with a single run over the 11 datasets and 10 values of the pair $(\alpha_b, \alpha_e)$ with $\alpha_b \in \{0.8, 0.9\}$ and $\alpha_e \in \{0.8, 0.85, 0.9, 0.95, 0.99\}$. The experiment with reward noise $\nu = 0.25$ is reported in the main paper (Figure 2), whereas the noiseless $\nu = 0$ (deterministic rewards) is provided in Figure 6. The results are in line with the stochastic case.

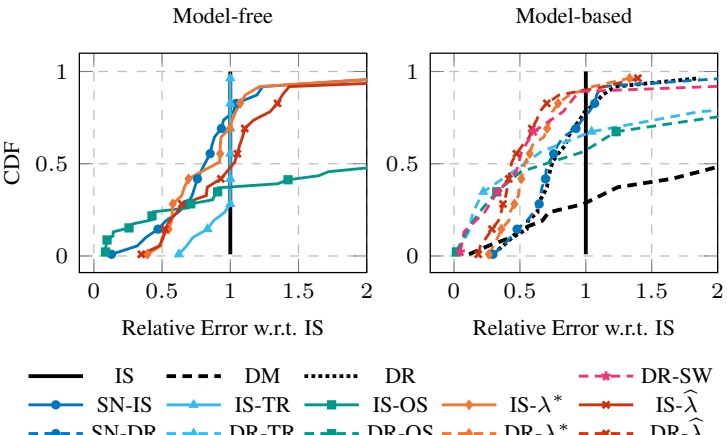

Figure 6: CDF of the absolute error normalized by IS error for deterministic rewards, across 110 conditions for model-free estimators (left) and model-based ones (right).

For the case of the *letter* dataset, we report the experiments with additional choices of $\alpha_e$ (10, and 11).

| Estimator / $n$ | 100 | 200 | 500 | 1000 | 2000 | 5000 | 10000 | 20000 |
|---|---|---|---|---|---|---|---|---|
| IS | **20.04±1.24** | **21.77±2.46** | 14.03±0.57 | 8.40±0.20 | 6.13±0.09 | 2.77±0.03 | 1.83±0.01 | 1.10±0.01 |
| SN-IS | 27.34±1.67 | 23.16±1.40 | 16.86±0.46 | 11.94±0.25 | 7.37±0.13 | 2.59±0.03 | 1.74±0.01 | 1.17±0.01 |
| IS-TR | **20.04±1.24** | **18.17±1.60** | 13.96±0.57 | 8.40±0.20 | 6.13±0.09 | 2.77±0.03 | 1.83±0.01 | 1.10±0.01 |
| IS-OS | 24.47±1.50 | 32.30±1.17 | 15.37±0.56 | 17.35±0.46 | 16.46±0.37 | 30.70±0.15 | 34.03±0.02 | 33.67±0.01 |
| IS-$\lambda$* | **20.48±1.33** | **16.77±1.14** | 10.06±0.34 | 6.61±0.16 | 5.30±0.07 | 2.88±0.03 | 2.08±0.01 | 1.16±0.01 |
| IS-$\lambda$** | **20.80±1.38** | **16.11±0.93** | 9.56±0.30 | 6.62±0.15 | 5.19±0.07 | 3.06±0.03 | 2.31±0.01 | 1.29±0.01 |
| IS-$\widehat{\lambda}$ | **22.60±1.52** | **17.06±0.75** | 10.22±0.28 | 7.77±0.16 | 5.61±0.08 | 3.32±0.03 | 2.50±0.02 | 1.37±0.01 |
| DM | 28.86±1.92 | 27.56±0.95 | 41.04±0.26 | 41.94±0.11 | 42.87±0.05 | 47.06±0.01 | 47.58±0.01 | 47.51±0.00 |
| DR | **26.54±4.51** | 25.56±2.43 | 16.69±0.72 | 9.12±0.20 | 5.62±0.09 | 2.14±0.02 | **1.25±0.01** | **0.83±0.00** |
| SN-DR | **25.62±3.21** | 24.87±1.79 | 18.94±0.62 | 12.36±0.23 | 7.19±0.12 | 2.46±0.02 | **1.57±0.01** | 1.07±0.01 |
| DR-TR | **18.97±1.12** | **16.54±0.70** | 20.95±0.23 | 17.93±0.09 | 17.90±0.06 | 22.73±0.01 | 23.45±0.01 | 23.18±0.00 |
| DR-OS | **18.87±1.18** | 19.21±0.55 | 17.15±0.38 | 12.01±0.23 | 8.67±0.11 | 17.04±0.06 | 17.88±0.02 | 18.49±0.02 |
| DR-SW | 23.97±1.28 | **16.66±1.13** | **4.58±0.18** | **4.64±0.09** | 4.76±0.05 | **0.75±0.01** | **1.31±0.01** | **0.77±0.00** |
| DR-$\lambda$* | **21.84±2.30** | **18.16±1.35** | 11.26±0.47 | 6.53±0.14 | **4.59±0.07** | 1.78±0.02 | **1.23±0.01** | **0.72±0.00** |
| DR-$\lambda$** | **21.00±2.01** | **16.70±1.18** | 10.00±0.41 | 5.82±0.13 | **4.27±0.06** | 1.67±0.02 | **1.24±0.01** | **0.69±0.00** |
| DR-$\widehat{\lambda}$ | **19.45±1.62** | **14.35±0.95** | 7.89±0.34 | **4.88±0.11** | **3.88±0.06** | 1.60±0.02 | **1.26±0.01** | **0.68±0.00** |

Table 10: Absolute error (multiplied by 100) in the *letter* dataset varying the number of samples $n$ for the different estimators, when $\alpha_b = 0.5$ and $\alpha_e = 0.99$ (mean $\pm$ std, 10 runs). For each column, the estimator with smallest absolute error and the ones not statistically significantly different from that one (Welch's t-test with $p < 0.05$) are in bold.

| Estimator / $n$ | 100 | 200 | 500 | 1000 | 2000 | 5000 | 10000 | 20000 |
|---|---|---|---|---|---|---|---|---|
| IS | **10.08±0.91** | **20.07±5.66** | 20.23±1.60 | 13.52±0.42 | 12.23±0.24 | 6.49±0.05 | 3.62±0.03 | 2.74±0.01 |
| SN-IS | 15.85±1.60 | 18.18±1.71 | 26.34±0.75 | 23.84±0.35 | 12.91±0.20 | 5.96±0.05 | 4.15±0.03 | 2.14±0.01 |
| IS-TR | **10.08±0.91** | **12.02±1.66** | 13.94±0.65 | 11.34±0.22 | 11.38±0.20 | 6.40±0.05 | 3.62±0.03 | 2.74±0.01 |
| IS-OS | 26.61±0.75 | 53.26±5.63 | 38.57±1.10 | 32.35±0.09 | 30.73±0.06 | 25.41±0.02 | 24.48±0.01 | 23.76±0.00 |
| IS-$\lambda$* | **10.17±0.91** | **12.03±1.62** | 13.11±0.52 | 10.33±0.13 | 8.89±0.09 | 3.88±0.03 | 2.74±0.02 | 2.35±0.01 |
| IS-$\lambda$** | **10.21±0.91** | **11.21±1.21** | 12.15±0.37 | 9.80±0.09 | 8.23±0.07 | 3.54±0.03 | 2.61±0.02 | 2.32±0.01 |
| IS-$\widehat{\lambda}$ | **11.00±0.93** | **9.73±0.37** | 9.72±0.18 | 8.95±0.09 | 7.82±0.07 | 3.67±0.03 | 2.98±0.02 | 2.56±0.01 |
| DM | 22.00±1.92 | **9.78±0.47** | **7.27±0.19** | **3.49±0.08** | **2.83±0.05** | 9.16±0.02 | 9.97±0.01 | 9.89±0.00 |
| DR | **35.50±15.08** | 33.18±7.27 | 30.82±1.83 | 24.87±0.82 | 14.16±0.27 | 6.19±0.06 | 3.46±0.03 | 1.48±0.01 |
| SN-DR | 19.47±3.41 | 20.94±2.62 | 24.69±1.07 | 21.41±0.43 | 13.66±0.16 | 6.41±0.06 | 3.57±0.03 | 1.57±0.01 |
| DR-TR | 12.26±1.14 | **10.90±0.61** | **6.65±0.23** | 9.98±0.09 | 10.29±0.04 | **2.34±0.01** | **0.95±0.00** | **0.60±0.00** |
| DR-OS | 12.57±1.23 | **8.59±0.49** | **6.73±0.27** | 5.25±0.13 | 9.05±0.09 | **2.60±0.01** | 1.97±0.01 | 1.21±0.00 |
| DR-SW | 12.46±1.09 | 11.73±0.64 | **7.52±0.25** | 11.31±0.10 | 11.59±0.04 | 3.49±0.01 | 2.09±0.00 | 1.69±0.00 |
| DR-$\lambda$* | 18.78±3.41 | 16.07±2.09 | 15.26±0.66 | 13.55±0.31 | 8.96±0.15 | 3.97±0.04 | 2.44±0.02 | 1.24±0.01 |
| DR-$\lambda$** | 17.51±2.59 | 14.39±1.59 | 13.17±0.52 | 11.89±0.24 | 8.22±0.14 | 3.55±0.03 | 2.22±0.02 | 1.21±0.01 |
| DR-$\widehat{\lambda}$ | 14.58±1.18 | **10.32±0.58** | 8.42±0.23 | 11.33±0.19 | 11.02±0.23 | 2.83±0.02 | 2.00±0.01 | 1.26±0.00 |

Table 11: Absolute error (multiplied by 100) in the *letter* dataset varying the number of samples $n$ for the different estimators, when $\alpha_b = 0.9$ and $\alpha_e = 0.99$ (mean $\pm$ std, 10 runs). For each column, the estimator with smallest absolute error and the ones not statistically significantly different from that one (Welch's t-test with $p < 0.05$) are in bold.

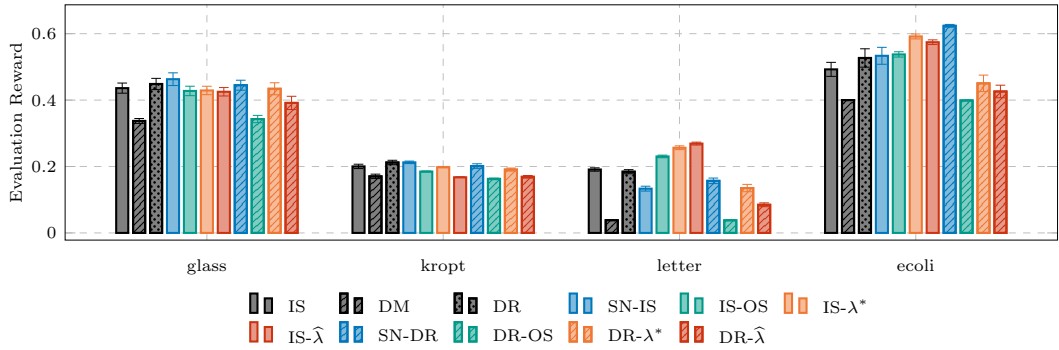

Figure 7: Evaluation reward for four different datasets after 1000 iterations (4000 iterations for *letter*) of gradient ascent with a Boltzmann policy and the non-regularized objective ($\zeta = 0$) (mean $\pm$ std, 10 runs).

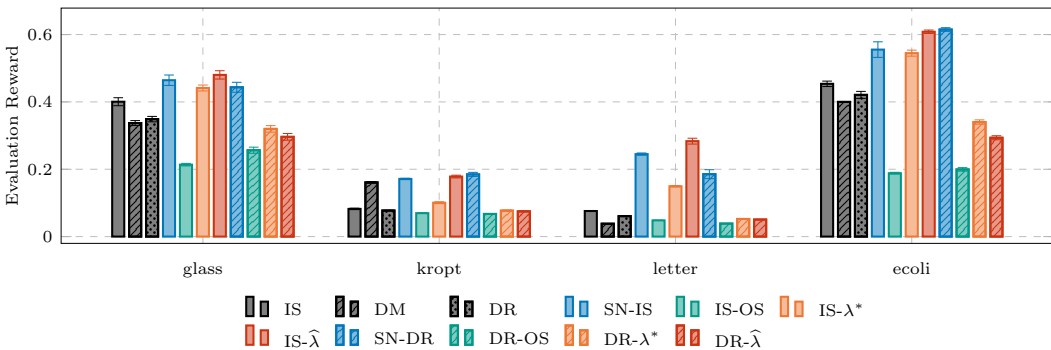

Figure 8: Evaluation reward for four different datasets after 1000 iterations (4000 iterations for *letter*) of gradient ascent with a Boltzmann policy and the regularized objective ($\zeta = 0.1$) (mean $\pm$ std, 10 runs).

## B.2 Off-Policy Learning

**Experimental Setting**   The optimization is performed by gradient ascent on the objective function:

$$\mathcal{L}(\boldsymbol{\theta}) = \widehat{v}(\pi_{\boldsymbol{\theta}}) - \frac{\zeta}{n} \sum_{i \in [n]} I_2(\pi_{\boldsymbol{\theta}}(\cdot | x_i) \| \pi_b(\cdot | x_i)),$$

where $\widehat{v}(\pi_{\boldsymbol{\theta}})$ is the estimated value function using the different estimators, that is a function of the target policy $\pi_{\boldsymbol{\theta}}$. The second term is the empirical average of the divergence between the target $\pi_{\boldsymbol{\theta}}$ and the behavioral policy $\pi_b$. The regularizer is controlled by the regularization parameter $\zeta \geqslant 0$. The gradient optimization is performed in mini-batch made of 32 samples and the learning rate is selected with RMSprop, with $0.05$ as base learning rate.

**Complete Results**   In Figure 7 and in Figure 8 we report the complete results, in the setting presented in the main paper, for the non-regularized ($\zeta = 0$) and the regularized objective ($\zeta = 0.1$) respectively. The experiments with the regularized objective are limited to *glass* and *ecoli* datasets. We report the corresponding learning curves for the non-regularized (Figure 9) and the regularized objectives (Figure 10).

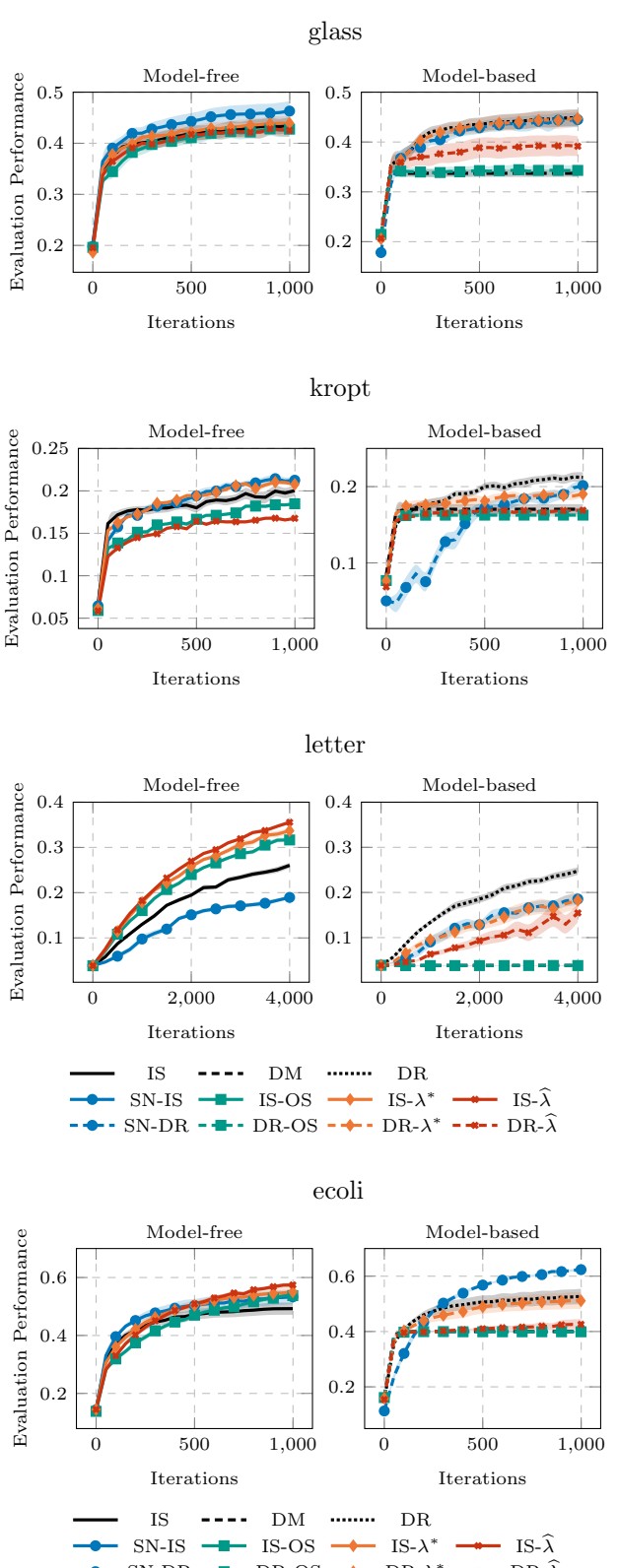

Figure 9: Evaluation reward for the four datasets comparing the learning curve of different estimators with the non-regularized objective ($\zeta = 0$) (mean $\pm$ std, 10 runs).

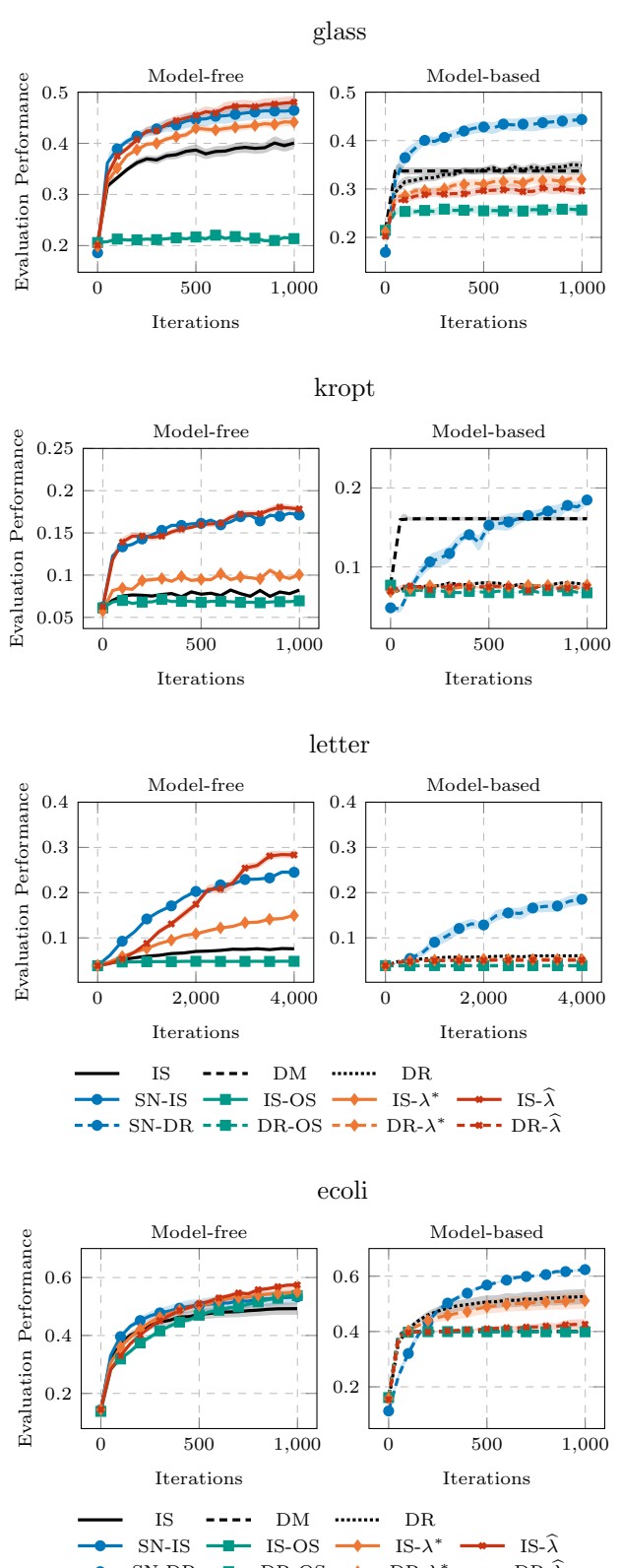

Figure 10: Evaluation reward for the four datasets comparing the learning curve of different estimators with the regularized objective ($\zeta = 0.1$) (mean $\pm$ std, 10 runs).

## C  Bound Comparison and Optimization

In this appendix, we provide a comparison between the bounds of Lemma A.3 and Theorem 5.1 and show how to numerically optimize the former. For the sake of simplicity, we restrict our attention to $\alpha = 2$ and we denote with $B^{**}(\lambda)$ the bound of Lemma A.3, with $\lambda^{**}$ its global minimum, with $B^*(\lambda)$ the bound of Theorem 5.1, and with $\lambda^*$ its global mimimum.

$B^{**}(\lambda)$ displays a pretty intricate dependence on $\lambda$ that is not easy to optimize. As we can notice from Figure 11, the bound based on the values of its terms admits either one or two local minima. In any case $\lambda = 1$ is a value of interest, leading to a bound of the form:

$$\hat{\mu}_{n,1} - \mu \leqslant \|f\|_\infty \sqrt{\frac{2\log\frac{1}{\delta}}{n}} + \frac{2\|f\|_\infty \log\frac{1}{\delta}}{3n} + \|f\|_\infty \sqrt{J_2(P\|Q)}.$$

In such a case, we are replacing the importance weight with the value of 1 and we are estimating the mean under the target distribution with the mean of the behavioral distribution, paying the whole bias $\sqrt{J_2(P\|Q)} = \sqrt{I_2(P\|Q) - 1}$. Clearly, this circumstance is convenient only when $n$ is sufficiently small.

The bound of Theorem 5.1 $B^*$ is looser compared with that of Lemma A.3 $B^{**}$. We can see in Figure 12 that bound of $B^*$ is convex and yeilds an optimal value of $\lambda^*$ that is smaller compared to the optimal value $\lambda^{**}$ of $B^{**}$.

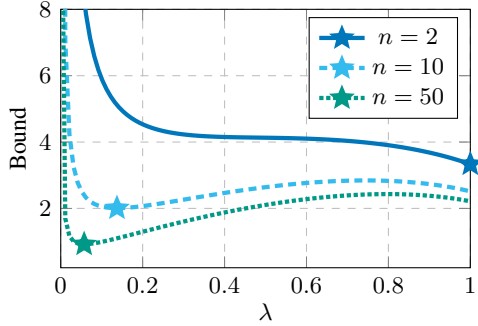

Figure 11: The bound of Lemma A.3 for $\alpha = 2$, $I_2(P\|Q) = 5$, $\delta = e^{-1}$, and $n \in \{2, 10, 50\}$. The minima are highlighted with the star.

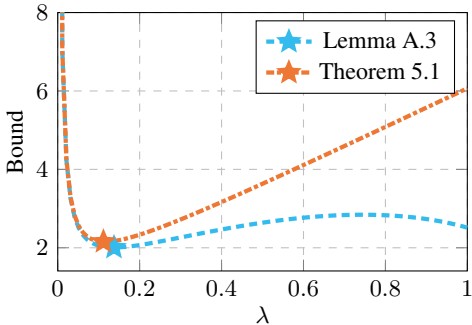

Figure 12: Comparison between the bounds of Lemma A.3 and Theorem 5.1 for $\alpha = 2$, $I_2(P\|Q) = 5$, $\delta = e^{-1}$, and $n = 10$. The minima are highlighted with the star.

### C.1  Numerical Optimization of the Bound of Lemma A.3

We now discuss how to find the global minimum of the bound presented in Lemma A.3 $B^{**}(\lambda)$. First of all, we observe that $B^{**}(\lambda)$ is continuously differentiable in $\lambda$:

$$\frac{\partial B^{**}(\lambda)}{\partial \lambda} = \sqrt{(I_2(P\|Q) - 1)((1-\lambda)I_2(P\|Q) + \lambda)} - \frac{2\log\frac{1}{\delta}}{3n\lambda^2}$$

---

**Algorithm 1** Root finding for bound $B^{**}$ of Lemma A.3

---

Compute the bound derivative $\frac{\partial B^{**}(\lambda)}{\partial \lambda}$

Apply Newton's method with $\lambda^*$ as initial guess obtaining $\lambda_1$ as numerical root (if exists)

**if** Newton's method failed to converge **or** $B^{**}(\lambda_1) < B(1)$ **then**

    **return** $1$

**else**

    **return** $\lambda_1$

**end if**

---

$$-\frac{(I_2(P\|Q)-1)\left(\sqrt{2\log\frac{1}{\delta}}+\lambda\sqrt{(I_2(P\|Q)-1)n}\right)}{2\sqrt{n((1-\lambda)I_2(P\|Q)+\lambda)}}.$$

We start proving that $\frac{\partial B^{**}(\lambda)}{\partial \lambda}$ is a strictly concave function of $\lambda$:

$$\frac{\partial^2}{\partial\lambda^2}\left(\frac{\partial B^{**}(\lambda)}{\partial\lambda}\right)=\frac{\partial^3 B^{**}(\lambda)}{\partial\lambda^3}=-\frac{4\log\frac{1}{\delta}}{n\lambda^4}-\frac{3(I_2(P\|Q-1)^{7/2}\lambda}{8((1-\lambda)I_2(P\|Q)+\lambda)^{5/2}}$$

$$-\frac{3(I_2(P\|Q)-1)^{5/2}}{4((1-\lambda)I_2(P\|Q)+\lambda)^{3/2}}-\frac{3(I_2(P\|Q)-1)^3\sqrt{\log\frac{1}{\delta}}}{4\sqrt{2n}((1-\lambda)I_2(P\|Q)+\lambda)^{5/2}}<0.$$

We now prove that $\frac{\partial B^{**}(\lambda)}{\partial \lambda}$ admits at most two roots. By contradiction, suppose $\frac{\partial B^{**}(\lambda)}{\partial \lambda}$ admits three roots $\lambda_1 < \lambda_2 < \lambda_3$. By Rolle's theorem, there must exist $\lambda_1 < \lambda_{12} < \lambda_2$ and $\lambda_2 < \lambda_{23} < \lambda_3$ such that $\frac{\partial^2 B^{**}(\lambda)}{\partial\lambda^2}(\lambda_{12})=\frac{\partial^2 B^{**}(\lambda)}{\partial\lambda^2}(\lambda_{23})=0$. Again, by Rolle's theorem, there must exist $\lambda_{12}<\lambda_{1223}<\lambda_{23}$ such that $\frac{\partial^3 B^{**}(\lambda)}{\partial\lambda^3}(\lambda_{1223})=0$, which is a contradiction being $\frac{\partial B^{**}(\lambda)}{\partial \lambda}$ concave. Thus we consider three cases:

- $\frac{\partial B^{**}(\lambda)}{\partial \lambda}$ admits no roots. It follows that the global minimum of $B^{**}$ is on the border $\{0,1\}$. Since $\lim_{\lambda\to 0^+} B^{**}(\lambda)=\infty$, the minimum is in $\lambda^{**}=1$.

- $\frac{\partial B^{**}(\lambda)}{\partial \lambda}$ admits one root. It is simple to prove that for sufficiently large $\lambda$ (possibly larger than 1, but this does not matter of the sake for the function study) we have $\frac{\partial B^{**}(\lambda)}{\partial \lambda}<0$. Being also $\lim_{\lambda\to 0^+}\frac{\partial B^{**}(\lambda)}{\partial \lambda}=-\infty$, we conclude that the root must be a saddle point and, consequently, $\lambda^{**}=1$.

- $\frac{\partial B^{**}(\lambda)}{\partial \lambda}$ admits two roots $\lambda_1 < \lambda_2$. Thus, there must exist $\lambda_1 < \lambda_{12} < \lambda_2$ such that $\frac{\partial^2 B^{**}(\lambda)}{\partial\lambda^2}(\lambda_{12})=0$. Since $\frac{\partial^2 B^{**}(\lambda)}{\partial\lambda^2}$ is non-increasing, being $\frac{\partial B^{**}(\lambda)}{\partial \lambda}$ concave, it must be that $\frac{\partial^2 B^{**}(\lambda)}{\partial\lambda^2}(\lambda_1)>0$ and $\frac{\partial^2 B^{**}(\lambda)}{\partial\lambda^2}(\lambda_2)<0$. Thus, $\lambda_1$ is a local minimum and $\lambda_2$ a local maximum. It follows that $\lambda^{**}\in\arg\min_{\lambda\in\{\lambda_1,1\}}B^{**}(\lambda)$.

Thus, based on the function study, it suffices to find numerically the smallest root $\lambda_1$ (whenever it exists) of $\frac{\partial B^{**}(\lambda)}{\partial \lambda}$ and compare its bound value $B^{**}(\lambda_1)$ with $B^{**}(1)$. This task can be carried out using numerical root finding, e.g., Newton's method, using as initial guess 0 or $\lambda^*$, having observed that in the optimal correction parameter $\lambda^*$ of the simplified bound $B^*$ the derivative $\frac{\partial B^{**}(\lambda)}{\partial \lambda}$ is negative. The procedure is summarized in Algorithm 1

# D Comparison of Estimators for CMABs

| Estimator | Formula |
|-----------|---------|
| Direct method (DM) | $\frac{1}{n}\sum_{i\in[n]}\sum_{a\in\mathcal{A}}\pi_e(a\|x_i)\widehat{r}(x_i,a)$ |
| Inverse propensity scoring (IPS) | $\frac{1}{n}\sum_{i\in[n]}\frac{\pi_e(a_i\|x_i)}{\pi_b(a_i\|x_i)}r_i$ |
| Doubly robust (DR) | $\frac{1}{n}\sum_{i\in[n]}\sum_{a\in\mathcal{A}}\pi_e(a\|x_i)\widehat{r}(x_i,a)+\frac{1}{n}\sum_{i\in[n]}\frac{\pi_e(a_i\|x_i)}{\pi_b(a_i\|x_i)}(r_i-\widehat{r}(x_i,a_i))$ |

Table 12: Overview of the classical off-policy estimators for CMABs. $\pi_b$ and $\pi_e$ denote the behavioral and target policies respectively and $\widehat{r}$ the estimated reward function.

# E Analysis of the IS-OS estimator

The IS-OS (optimistic shrinkage) [56] is based on the weight transformation:

$$\omega_\tau^{\mathrm{OS}}(y)=\frac{\tau\omega(y)}{\omega(y)^2+\tau}.$$

First of all, we notice that when $P=Q$ a.s. the weight becomes $\omega_\tau^{\mathrm{OS}}(y)=\frac{\tau}{\tau+1}$, so the estimator is biased. We start by observing that the corrected weight $\omega_\tau^{\mathrm{OS}}(y)$ converges to zero when the non-corrected weight is either zero or infinity. Thus, the maximum value of the weight must be in between. We compute it by vanishing the derivative:

$$\frac{\partial}{\partial\omega}\frac{\tau\omega}{\omega^2+\tau}=0 \quad\Longrightarrow\quad \omega=\sqrt{\tau}.$$

From which, we obtain the maximum value of the weight equal to $\frac{\sqrt{\tau}}{2}$. We now focus on the following result concerning the bias and the variance of the IS-OS estimator.

**Lemma E.1.** *Let $P,Q\in\mathscr{P}(\mathcal{Y})$ be two probability distributions with $P\ll Q$. For every $\tau\geqslant 0$, the bias and variance of the IS-OS estimator can be bounded as:*

$$\left|\mathbb{E}_{y\sim Q}\left[\widehat{\mu}_{n,\tau}^{OS}\right]-\mu\right|\leqslant\frac{\|f\|_\infty}{\tau}I_3(P\|Q), \qquad \operatorname*{\mathbb{V}ar}_{y_i\sim Q}\left[\widehat{\mu}_{n,\tau}^{OS}\right]\leqslant\frac{\|f\|_\infty^2}{n}I_2(P\|Q).$$

*Proof.* Let us start with the bias. Based also on Lemma A.1, we consider the following inequality:

$$\mathbb{E}_{y\sim Q}\left[\left|\omega_\tau^{\mathrm{OS}}(y)-\omega(y)\right|\right]=\mathbb{E}_{y\sim Q}\left[\frac{\omega(y)^3}{\omega(y)^2+\tau}\right]\leqslant\mathbb{E}_{y\sim Q}\left[\frac{\omega(y)^3}{\tau}\right]=\frac{I_3(P\|Q)}{\tau}.$$

We consider now the variance term and derive a bound on the second moment of the OS weight:

$$\mathbb{E}_{y\sim Q}\left[\omega_\tau^{\mathrm{OS}}(y)^2\right]=\mathbb{E}_{y\sim Q}\left[\left(\frac{\omega(y)\tau}{\omega(y)^2+\tau}\right)^2\right]\leqslant\mathbb{E}_{y\sim Q}\left[\omega(y)^2\right]=I_2(P\|Q).$$

$\square$

We now move to the concentration result.

**Theorem E.1.** *Let $P,Q\in\mathscr{P}(\mathcal{Y})$ be two probability distributions with $P\ll Q$. Then, having selected $\tau^*=\left(\frac{6nI_3(P\|Q)}{\log\frac{1}{\delta}}\right)^{\frac{2}{3}}$ for the IS-OS estimator, for every $\delta\in[0,1]$, with probability at least $1-\delta$ it holds that:*

$$\widehat{\mu}_{n,\tau}^{OS}-\mu\leqslant\|f\|_\infty\sqrt{\frac{2I_2(P\|Q)\log\frac{1}{\delta}}{n}}+\|f\|_\infty\sqrt[3]{\frac{3I_3(P\|Q)\left(\log\frac{1}{\delta}\right)^2}{4n^2}}.$$

*Proof.* We apply Bernstein's inequality to the estimator, staring for the bias and variance bounds of Lemma E.1:

$$\widehat{\mu}_{n,\tau}^{\text{OS}} - \mu = \widehat{\mu}_{n,\tau}^{\text{OS}} - \mathop{\mathbb{E}}_{y_i \sim Q}[\widehat{\mu}_{n,\tau}^{\text{OS}}] + \mathop{\mathbb{E}}_{y_i \sim Q}[\widehat{\mu}_{n,\tau}^{\text{OS}}] - \mu$$

$$\leqslant \|f\|_\infty \sqrt{\frac{2I_2(P\|Q)\log\frac{1}{\delta}}{n}} + \frac{\|f\|_\infty \sqrt{\tau}\log\frac{1}{\delta}}{3n} + \frac{\|f\|_\infty}{\tau} I_3(P\|Q).$$

We now minimize the bound as a function of $\tau$ by vanishing the derivative to obtain:

$$\tau^* = \left(\frac{6nI_3(P\|Q)}{\log\frac{1}{\delta}}\right)^{\frac{2}{3}}.$$

By substituting $\tau^*$ we obtain the result. $\qquad\square$