# OpenReview forum: "Subgaussian and Differentiable Importance Sampling for Off-Policy Evaluation and Learning"
_NeurIPS.cc/2021/Conference — NeurIPS 2021 Spotlight_

### Official Review · Reviewer_ZK8F · 2021-06-29

**Rating:** 8
**Confidence:** 2

**Summary:**

The paper addresses off-policy evaluation. It proposes to transform importance weights using the function $\frac{\omega(y)}{1-\lambda + \lambda \omega(y)}$, where $\omega(y) = p(y)/q(y)$ is the standard importance weight. The estimator which uses these transformed weights is analyzed both theoretically and empirically. The obtained estimator is unique in having both subgaussian concentration rate and being differentiable. The estimator depends on the hyperparameter $\lambda$, for which the authors give an estimation procedure that only depends on observable quantities.

**Limitations And Societal Impact:**

The authors do not discuss societal impact, arguing that the paper is theoretical. I agree with that assessment: off-policy evaluation can be misused but the potential for such misuse is same for this paper as for any other research on off-policy methods.

**Main Review:**

The paper is original and high-quality.  It is clearly written - I feel I understand the main message even though I am a practitioner and do not come from this subcommunity. The contribution is significant and well within the scope of NeurIPS.

As a practitioner, I find it hard to comment on the proofs, but I have a couple of questions to the authors.
1. You introduce the importance weight transformation (your definition 4.1) for arbitrary values $s$ but then the paper focusses on using $s=-1$ only. Why is that? I feel the paper should either use $s\neq-1$ at least once in practice or else not discuss it.
2. You address the contextual bandit setting. How hard would it be to generalise the result to off-policy evaluation for MDPs? Do you have a hunch about whether the same weight transformation would work? Of course, I am not expecting a proof (the paper already has a lot of material), but a couple of sentences of discussion would be nice.
3. In the contextual bandit comprehensive experiment (as described in section B.1.2 in the appendix), you tried two values of alpha for the behavioral policy: $\alpha_b \in \{ 0.8, 0.9\}$. Was there a particular reason you did not try more extreme values (closer to one)? Such evaluation would be useful because it makes the task harder (the regular, untransformed importance weights can become quite large).
4. In Table 4, the most successful approaches for large n are those that plug the proposed weight transform into the doubly robust estimator. However, theorems 5.1 and 6.1 talk about "regular" IS, not about doubly robust estimation. Is there a theoretical justification for combining doubly robust estimation with the proposed importance weight transform?

Very minor points:
- Line 81: "more in general" => "more generally"
- Line 120: "result" => "results"
- Line 772 in the appendix: missing word, probably "samples"

**Time Spent Reviewing:**

5

---

> ### Author Response · Authors · 2021-08-09
> **Response to Reviewer ZK8F**
>
> We thank the reviewer for the positive feedback. We address the questions in the following:
>
> 1. We decided to present the general formulation as power mean with exponent $s$ to show that the class of smooth transformations is wider than the case $s=-1$, which instead we study in detail. The reasons why selecting $s=-1$, as we mentioned in Footnote 4, are two: (i) it allows achieving subgaussian concentration; (ii) the analytical form of the corrected weight is convenient for the theoretical analysis (differently from a general value of $s$). See also the response to Reviewer 8Vhk for more hints on the theoretical properties of the cases $s \neq 1$.
>
> As the reviewer suggested, we tested empirically different values of $s$, in the same setting of Table 3 with $n=500$. Since for general $s$, we do not have a principled way to select $\lambda$, we consider different values of $\lambda$. The results are reported in the following table (Welch’s t-test with $p \le 0.05$). We can see that the best results are obtained with $s\in\\{-1,-2\\}$. We will add the experiment in the final version.
>
> $$
> \begin{array}{|c|cccc|}\hline
> s \backslash \lambda & 0 & 0.1 & 0.2 & 0.5 \\\\ \hline
> -\infty & 3.12 \pm 0.29 & 3.12 \pm 0.29 & 3.12 \pm 0.29 & 3.12 \pm 0.29 \\\\
> -5 & 6.73 \pm 1.21 & \mathbf{2.70 \pm 0.30} & 2.77 \pm 0.30 & 2.57 \pm 0.31 \\\\
> -2 & 6.73 \pm 1.21 & \mathbf{2.45 \pm 0.34} & \mathbf{2.42 \pm 0.32} & \mathbf{2.28 \pm 0.32} \\\\
> -1 & 6.73 \pm 1.21 & \mathbf{2.72 \pm 0.47} & \mathbf{2.47 \pm 0.37} & \mathbf{2.18 \pm 0.32} \\\\
> -0.5 & 6.73 \pm 1.21 & 3.44 \pm 0.64 & \mathbf{2.71 \pm 0.47} & \mathbf{2.20 \pm 0.34} \\\\
> 0 & 6.73 \pm 1.21 & 4.83 \pm 0.89 & 3.66 \pm 0.68 & \mathbf{2.38 \pm 0.38} \\\\
> 0.5 & 6.73 \pm 1.21 & 5.69 \pm 1.05 & 4.85 \pm 0.89 & 3.03 \pm 0.52 \\\\ \hline
> \end{array}
> $$
>
>
> 2. When moving to MDPs, one way to deal with off-policy evaluation is to consider the product of policy ratios in order to define the importance weight over the trajectory $\omega(\tau) = \prod_{t=1}^T \frac{\pi_e(a_t|s_t)}{\pi_b(a_t|s_t)}$. We think that the correction could be applied in two ways. First, we can directly correct the full product $\omega(\tau)$ using a single value of $\lambda$. This would disregard the property of the underlying MDP. Second, we can apply the correction to the single ratios $\frac{\pi_e(a_t|s_t)}{\pi_b(a_t|s_t)}$ with $T$ values $\lambda_t$ possibly different for every $t$. This latter approach might be more flexible, allowing for different correction intensities at different steps. We will include this discussion in the paper.
>
> 3. We showed in the paper the setting $\alpha_b \in\\{0.8, 0.9\\}$ and $\alpha_e \in \\{0.8, 0.85, 0.9, 0.95, 0.99\\}$ since we wanted to simulate a realistic scenario in which the behavioral policy (defined via $\alpha_b$) is usually more stochastic (explorative) compared to the target policy (defined via $\alpha_e$), similarly to the experimental setting of [3]. We also tested higher values of $\alpha_b$ (specifically, $\alpha_b \in \\{0.95, 0.99 \\}$), confirming the results presented in the paper, although the advantage of our corrected estimators becomes progressively less visible since, with more extreme values of $\alpha_b$, the samples from the behavioral policies tend to provide too little information about the target policies.
>
> 4. We look at the experiments with doubly robust estimator as an application of our weight transformation to a class of well-known estimators. Intuitively, we can say that our weight transformation brings benefits since it mitigates the degeneracy of the importance weights that are employed by the doubly robust estimator. From the theoretical standpoint, we believe it would be challenging to provide formal results since it would depend on the properties (risk bounds) of the regressor employed to estimate the reward function.
>
> Thank you also for pointing out the typos.
>
> [3] Su, Yi, Maria Dimakopoulou, Akshay Krishnamurthy, and Miroslav Dudík. "Doubly robust off-policy evaluation with shrinkage." In International Conference on Machine Learning, pp. 9167-9176. PMLR, 2020.

---

### Official Review · Reviewer_4xpm · 2021-07-17

**Rating:** 8
**Confidence:** 3

**Summary:**

This paper considers a power mean correction term for off-policy evaluation and learning. The resulting estimator has a subgaussian concentration rate, improving on other variants of importance sampling (which has a polynomial concentration rate). The authors bound the variance and bias of the resulting estimator. These bounds are functions of a tuned parameter $\lambda$, where higher values of lambda increase the bias but reduce the variance giving practitioners the ability to tune the properties to their problem. The estimator also has other beneficial properties: the correction term is differentiable, and the free parameter $\lambda$ can be tuned through data.

**Limitations And Societal Impact:**

I believe the authors do a good job detailing their algorithms. This is an algorithmic paper, so there is little to not in societal impact.

**Main Review:**

Overall, I think the paper is well written and provides a really interesting algorithm with interesting analysis of the new estimator and of vanilla importance-sampling. The work is well situated in the literature, although my breadth of knowledge of the CMAB literature is limited. I've also lightly checked parts of the theory for correctness, but have not thoroughly checked all the proofs. While the theoretical sections are quite dense, I think the authors do a good job giving readers the appropriate context to wade through the theory. The empirical comparisons also seem quite robust and provide comparisons to several baselines, which has been expanded from prior submissions (as they report). I have very few comments overall, and think the paper is a strong accept assuming there are no significant errors in the theory.

- More runs would make the empirical results better.
- Line 637: I think you mean just $\leq$ instead of just $\leq =$ in the equation.

**Time Spent Reviewing:**

4

---

> ### Author Response · Authors · 2021-08-09
> **Response to Reviewer 4xpm**
>
> We thank the reviewer for the positive feedback. We are glad that the paper was appreciated. We will increase the number of runs from 10 to 50 for the plots of Table 4, Figures 3 and 4. Thank you also for reporting the typo.

---

### Official Review · Reviewer_9nhQ · 2021-07-19

**Rating:** 7
**Confidence:** 3

**Summary:**

This paper proposes an off-policy evaluation estimator that is sub-Gaussian and differentiable. They provide an anti-concentration bound of the vanilla inverse propensity score weighting estimator, which implies that it is not sub-Gaussian. They propose a correction term on the inverse propensity score that shrinks the score. They show that the proposed estimator is sub-Gaussian under some conditions, and is differentiable with respect to the target policy parameters. They provide some empirical study to show that the proposed estimator is competitive with existing estimators.

**Limitations And Societal Impact:**

yes

**Main Review:**

The paper is well-structured and easy to follow.

The theoretical study of the concentration property of off-policy evaluation estimators could help better understand the behavior of these estimators. I like Table 2 in the paper which summarizes some theoretical properties of some model-free estimators. The paper presents new anti-concentration bound on the vanilla IPS estimator, and they show their proposed estimator is sub-Gaussian.

The data-driven approach for selecting \lambda makes the proposed estimator more applicable, and is theoretically grounded by concentration bound in the paper.

Since the paper is focused on concentration behaviors, I am wondering if the paper could show their concentration behaviors. I am curious whether IS, SN-IS, IS-OS are heavy-tailed empirically.

It would be great if the paper could discuss the concentration property of the model-based estimators, since it seems that these estimators perform much better than model-free estimators in both evaluation and learning. In particular, the discussion of the effect of the correction weight on the doubly robust estimator seems missing in the paper.

**Time Spent Reviewing:**

4

---

> ### Author Response · Authors · 2021-08-09
> **Response to Reviewer 9nhQ**
>
> We thank the reviewer for the insightful comments. We address the questions in the following.
>
> * (**Heavy-tailed behavior**) IS displays a heavy-tailed behavior as proved in [2] (Proposition C.2) for Gaussian distributions showing that the p.d.f. of the importance weight approaches zero with a polynomial rate $\widetilde{O}(y^{-1-c})$ for some constant $c>0$ depending on the variance of the two distributions. Instead, SN-IS and IS-OS are not heavy-tailed since the corresponding estimators are bounded in absolute value (see the “Maximum” column in Table 2). Nevertheless, they fail to achieve subgaussian concentration for general distributions.
>
> [2] Metelli, Alberto Maria, Matteo Papini, Francesco Faccio, and Marcello Restelli. "Policy optimization via importance sampling." In Proceedings of the 32nd International Conference on Neural Information Processing Systems, pp. 5447-5459. 2018.
>
> * (**Concentration of Model-based Estimators**) We did not study the properties of the model-based estimators because their concentration would depend on the estimated reward function. Such an analysis would require considering the risk bounds for the regressor employed in the reward function estimation (logistic regression in our case), and we believe this is out of the scope of the present paper. Instead, we look at the experiments with model-based estimators as an *application* of our importance-weight transformation. In this sense, the good performance of the model-based estimators can be explained as follows. First, model-based estimators, especially the doubly robust estimator, tend to outperform the model-free ones even without any weight correction. Second, our weight correction allows mitigating the residual variance effects of importance weighting. We will clarify this point in the paper.

---

> > ### Comment · Reviewer_9nhQ · 2021-08-28
> > **It would be great to show that Subgaussian concentration is important**
> >
> > Thank the authors for the response. I still have the concern that the paper does not show that concentration property is important empirically. The paper believes that concentration property for off-policy estimators is important, and shows that the proposed correction term can make the vanilla IPS estimator enjoy Subgaussian concentration. However, in the empirical evaluation, the ones that perform the best are model-based estimators, which are not shown to have concentration properties.
> >
> > If characterizing the concentration behaviors of the model-based estimators is hard, it might be helpful to empirically demonstrate their concentration behavior. If they have an empirical Subgaussian concentration, then it might show that Subgaussian concentration is truly helpful.
> >
> > For this reason, I could not raise my evaluation.

---

> > > ### Author Response · Authors · 2021-08-29
> > > **Re: It would be great to show that Subgaussian concentration is important**
> > >
> > > We take the opportunity to provide further clarification on the point. The general structure of a model-based estimator is the following:
> > > $$
> > > \begin{aligned}
> > > & \\underbrace{ \\frac{1}{n}\\sum_{i \\in [n]}\\sum_{a \\in \\mathcal{A}} \\pi_e(a|x_i) \\widehat{r}(x_i,a)} + \\underbrace{\\frac{1}{n} \sum_{i \\in [n]} \\omega(x_i,a_i) (r_i - \\widehat{r}(x_i,a_i))},\\\\
> > > & \\qquad\\qquad\\quad \\text{(i)} \\qquad\\qquad\\qquad\\qquad\\qquad \\qquad \\text{(ii)}
> > > \end{aligned}
> > > $$
> > > where $\widehat{r}$ is the estimated reward function and $\omega$ is a (possibly corrected) importance weight. The concentration properties of the model-based estimator depend on two factors: the concentration of the estimated reward function $\widehat{r}$ (i); the concentration of the IS-based part of the estimator (ii). (i) depends on the regressor (e.g., logistic regression), which, in our experiments, is the same for all the estimators. Instead, (ii) inherits the concentration properties of the importance weight $\omega$ employed (e.g., vanilla, self-normalized, our correction). In particular, given $\widehat{r}$, the DR estimator uses vanilla IS and, thus, is not subgaussian limitedly to part (ii); whereas DR-$\lambda^*$, using our corrected weight, is subgaussian for part (ii).
> > >
> > > We believe that the experiments provide evidence that the subgaussian property is helpful, even for the model-based estimators. Indeed, while the model-based estimators overall outperform the model-free ones, we note that, within the class of model-based estimators, our correction, that makes part (ii) of the estimators subgaussian, is beneficial. Indeed, referring to Figure 2 and Table 4, we see that DR-$\lambda^*$ outperforms DR, especially when the number of samples is small.

---

> > > > ### Comment · Reviewer_9nhQ · 2021-08-30
> > > > **Thanks for the response.**
> > > >
> > > > I still do not think that the empirical analysis sufficiently shows that Subgaussian is beneficial. Making part of an estimator Subgaussian does not make the whole estimator Subgaussian. We can always divide an estimator into two components where one is Subgaussian.
> > > >
> > > > What I would suggest is to conduct the estimation problem using different estimators multiple times, and see how heavy-tailed they are empirically. If the suggested corrections are "less heavy-tailed" than vanilla weights (even for model-based estimators), then it would empirically show that subgaussian concentration is beneficial.
> > > >
> > > > To be clear, I have no problem with the theory of the paper and I think it is complete and good without the analysis of model-based estimators. My point is that the empirical analysis could be improved.

---

> > > > > ### Author Response · Authors · 2021-08-31
> > > > > **Re: Thanks for the response.**
> > > > >
> > > > > We thank the reviewer for the suggestion on the experiment. We run $5000$ estimation processes using the "glass" dataset, $n=30$, $\alpha_e=0.999$, and $\alpha_b=0.9$. To compare the behavior between vanilla weights and our correction (for both model-free and model-based estimators) in terms of tail behavior, we considered the absolute error random variable $E$ (as in Table 4) and we estimated the *complementary cumulative distribution* $\Pr(E > \tau)$. Thus, for large values of $\tau$, the larger $\Pr(E>\tau)$, the heavier the tail, since more probability mass accumulates on the right of $\tau$. The following table reports the results for both model-based and model-free estimators:
> > > > > $$
> > > > > \begin{array}{|c|cccc|}\hline
> > > > > \text{Estimator} \backslash \tau & 10 & 20 & 50 & 100 & 200 & 500 & 1000 \\\\ \hline
> > > > > \text{IS} & 0.6742 & 0.5414 & 0.1754 & 0.0326 & 0.0326& 0.0198 &  0.0014\\\\
> > > > > \text{IS-}\lambda^* & 0.686 & 0.5416 & 0.176 & 0.0228 & 0.0056 & 0 & 0 \\\\ \hline
> > > > > \text{DR} & 0.6522 & 0.4094 & 0.117 & 0.0484 & 0.0378 & 0.0218 & 0.0022 \\\\
> > > > > \text{DR-}\lambda^* & 0.65 & 0.4046 & 0.1088 & 0.0362 & 0.009 & 0 & 0\\\\ \hline
> > > > > \end{array}
> > > > > $$
> > > > >
> > > > > We can see that, for $\tau > 50$, our corrected estimators (both model-free $\text{IS-}\lambda^* $ and model-based $\text{DR-}\lambda^*$) consistently display a lighter tail compared to the ones employing vanilla weights ($\text{IS}$ and $\text{DR}$).

---

> > > > > > ### Comment · Reviewer_9nhQ · 2021-08-31
> > > > > > **Thanks for the timely response**
> > > > > >
> > > > > > Thank you for your timely response and the experiments address my concern. I would suggest to include such results in the paper. It would be really helpful for the audience to see that the correction term empirically makes the estimation concentrate better (especially for the model-based ones). I would like to raise my evaluation to 7.

---

> > > > > > > ### Author Response · Authors · 2021-09-01
> > > > > > > **Thank you**
> > > > > > >
> > > > > > > Thank you! We will surely add these results in the final version.

---

### Official Review · Reviewer_8Vhk · 2021-07-20

**Rating:** 7
**Confidence:** 2

**Summary:**

This paper proposes a new estimator for off-policy evaluation and learning: the estimator modifies original importance sampling by adding a weight correction term controlled by one additional hyper-parameter. The paper analyzed the theoretical properties of the proposed estimator and used experiments to show the effectiveness of the proposed approach.

**Limitations And Societal Impact:**

The paper address the limitations and societal impact well.

**Main Review:**

+ The proposed estimator in harmonic correction (s=-1) is novel and simple its design.
+ The paper is well organized and easy to follow overall
+ The theoretical property of the harmonic correction part is sound and clearly discussed
- There are a lot of different variations of the power-mean correction of IS dicussed in Sec 4. I am wondering what properties they might have. If only harmonic mean version achieves subgaussian rate, and all later experiments only use harmonic mean version, this part feels a little disconnected to the main text of the paper.
- About the bound in Theorem 5.1 and Eq. 2, I am wondering this seems only to upper bound the difference, rather than the absolute difference. If the left hand side difference is smaller than zero, and the right hand side is positive, the bound holds automatically. Could it be possible that the proposed estimator always underestimates the $\mu$, thus always achieve the proposed bound? In my opinion, this bound doesn't help too much in our understanding.
- The motivation of subgaussian concentration is not very obvious for readers without enough context. I think the paper would benefit a lot by showing the practical/ theoretical benefit of achieving a subgaussian concentration bound.

**Time Spent Reviewing:**

3

---

> ### Author Response · Authors · 2021-08-09
> **Response to Reviewer 8Vhk**
>
> We thank the reviewer for the constructive comments. We address the last three points in the following:
>
> * (**Values of $s$ different from $-1$**) We opted to present the estimator in the general form of power-mean to give a more comprehensive idea of the kind of smooth transformations that can be implemented. We discuss the boundedness of the transformed weights, and consequently, the boundedness of the estimator in Lemma 4.1 for a general choice of $s$. As noted in Footnote 4, the choice $s=-1$ leads to a form of the estimator that is particularly convenient for theoretical analysis and already achieves the properties we are interested in (subgaussian concentration and differentiability). Surely, for $s \ge 1$ the tails of the estimator are even heavier and, therefore, no subgaussian concentration can be obtained. Instead, for $s < 1$ subgaussian concentration could be in principle achieved, although a probably more challenging analysis would be needed. See also response to Reviewer ZK8F for an experimental comparison of different values of $s$.
>
> * (**One-sided Bound vs Two-sided Bound**) We believe that it is not the case that the proposed estimator always underestimates $\mu$. Indeed, the result in Theorem 5.1 and in Eq. (2) is presented as a one-sided bound just for simplicity but actually holds in both directions. It is sufficient to instantiate Theorem 5.1 with the function $-f$ instead of $f$ (no constraint on the sign of $f$ is required) to get the reversed one-sided bound. More formally, let $\mu(f)$ and $\widehat{\mu}(f)$ be the true mean and the estimator using $f$ as function. First of all, we observe that $\mu(-f) = -\mu(f)$ and $\widehat{\mu}(-f) = -\widehat{\mu}(f)$. Therefore, from Theorem 5.1, having denoted with $B$ the right hand side, we can state: $\widehat{\mu}(-f) - \mu(-f) = -\widehat{\mu}(f) + \mu(f) \le B$, which in turn is equivalent to $\widehat{\mu}(f) - \mu(f) \ge -B$, that is the reversed bound. Combining this bound with Theorem 5.1 instantiated with $f$, we get the two-sided bound (on the absolute difference $|\widehat{\mu}(f) - \mu(f)|$), holding with probability $1-\delta/2$. We will clarify the point in the paper.
>
> * (**Motivation of Subgaussian Concentration**) The motivation behind the usage of subgaussian concentration instead of, for instance, Mean Square Error (MSE) bounds is that the latter tends to hide the heavy-tailed properties of the estimator, which is the case that occurs with importance sampling for general distributions. As noted in [1], MSE $\ll$ does not necessarily reflect the “typical” behavior of the error $\gg$. Instead, high-probability bounds, like subgaussian concentration bounds, allow highlighting these properties and are stronger compared with MSE. Moreover, from the high-probability bound, it is possible to derive an MSE bound. We will clarify this point in the paper.
>
> [1] Lugosi, Gábor, and Shahar Mendelson. "Mean estimation and regression under heavy-tailed distributions: A survey." Foundations of Computational Mathematics 19, no. 5 (2019): 1145-1190.

---

> > ### Author Response · Authors · 2021-09-01
> > **Post Rebuttal**
> >
> > We just wanted to check whether the rebuttal clarified the issues raised in the review. We are happy to engage further in case of additional questions.

---

> > > ### Comment · Area_Chair_cyHW · 2021-09-01
> > > **Message received**
> > >
> > > Your clarifications have been received and read. Thank you.

---

### Decision · Program_Chairs · 2021-09-27

**Decision:**

Accept (Spotlight)

**Comment:**

Importance sampling (IS, aka IPS) is an oft used and oft studied technique for counterfactual reasoning. While it is unbiased (under a "full support" assumption), it can have very large variance when the target policy's distribution is substantially different from that of the data collection (i.e., logging) policy, and this can have an adverse effect on policy evaluation and learning. This paper analyzes the concentration properties of the IS estimator, proving a new "anti-concentration bound," which shows that reward estimated by IS can be bounded away from the true expected reward with probability $\geq \delta$. The paper then proposes a power-mean transformation of the importance weights, and proves that the resulting estimator has subgaussian concentration. Experiments with the new estimator round out the paper, and the results show that the transformed importance weights generally outperform the "vanilla" importance weights.

The reviews agree that this is an interesting, novel analysis, and that it is well written and easy to follow. There were some concerns, but they have mostly been answered by the authors' responses. One unresolved comment (from Reviews 8Vhk and 9nhQ) is that the case for subgaussianity would be more convincing with more empirical evidence that it helps in practice. The reviewers were puzzled why the model-based baselines, which don't exhibit subgaussian tails, outperformed the proposed methods in some experiments.

Regardless, this is a strong paper that deserves to be accepted. Based on the novelty and relevance of the topic, I would go as far as to nominate it for a spotlight talk.